# Do We Really Need Complicated Model Architectures For Temporal Networks?

**Weilin Cong**
Penn State
weilin@psu.edu

**Si Zhang**
Meta
sizhang@meta.com

**Jian Kang**
University of Illinois at Urbana-Champaign
jiank2@illinois.edu

**Baichuan Yuan & Hao Wu & Xin Zhou**
Meta
{bcyuan,haowu1,markzhou}@meta.com

**Hanghang Tong**
University of Illinois at Urbana-Champaign
htong@illinois.edu

**Mehrdad Mahdavi**
Penn State
mzm616@psu.edu

## Abstract

Recurrent neural network (RNN) and self-attention mechanism (SAM) are the de facto methods to extract spatial-temporal information for temporal graph learning. Interestingly, we found that although both RNN and SAM could lead to a good performance, in practice neither of them is always necessary. In this paper, we propose GraphMixer, a conceptually and technically simple architecture that consists of three components: ① a *link-encoder* that is only based on multi-layer perceptrons (MLP) to summarize the information from temporal links, ② a *node-encoder* that is only based on neighbor mean-pooling to summarize node information, and ③ an MLP-based *link classifier* that performs link prediction based on the outputs of the encoders. Despite its simplicity, GraphMixer attains an outstanding performance on temporal link prediction benchmarks with faster convergence and better generalization performance. These results motivate us to rethink the importance of simpler model architecture. [Code].

## 1 Introduction

In recent years, temporal graph learning has been recognized as an important machine learning problem and has become the cornerstone behind a wealth of high-impact applications Yu et al. (2018); Bui et al. (2021); Kazemi et al. (2020); Zhou et al. (2020); Cong et al. (2021b). Temporal link prediction is one of the classic downstream tasks which focuses on predicting the future interactions among nodes. For example, in an ads ranking system, the user-ad clicks can be modeled as a temporal bipartite graph whose nodes represent users and ads, and links are associated with timestamps indicating when users click ads. Link prediction between them can be used to predict whether a user will click an ad. Designing graph learning models that can capture node evolutionary patterns and accurately predict future links is a crucial direction for many real-world recommender systems.

In temporal graph learning, recurrent neural network (RNN) and self-attention mechanism (SAM) have become the de facto standard for temporal graph learning Kumar et al. (2019); Sankar et al. (2020); Xu et al. (2020); Rossi et al. (2020); Wang et al. (2020), and the majority of the existing works focus on designing neural architectures with one of them and additional components to learn representations from raw data. Although powerful, these methods are conceptually and technically complicated with advanced model architectures. It is non-trivial to understand which parts of the model design truly contribute to its success, and whether these components are indispensable. Thus, in this paper, we aim at answering the following two questions:

Q1: Are RNN and SAM always indispensable for temporal graph learning? To answer this question, we propose GraphMixer, a simple architecture based entirely on the multi-layer perceptrons (MLPs) and neighbor mean-pooling, which does not utilize any RNN or SAM in its model architecture (Section 3). Despite its simplicity, GraphMixer could obtain outstanding results when comparing it

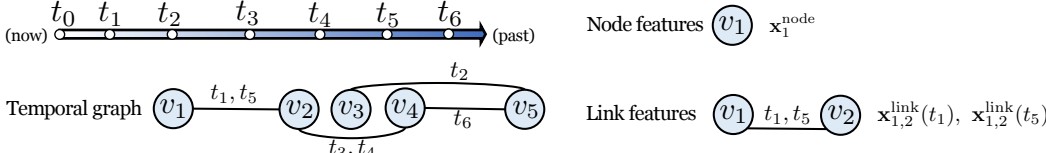

Figure 1: (Left) Temporal graph with nodes $v_1, \ldots, v_5$, per-link timestamps $t_1, \ldots, t_6$ indicate when two nodes interact. For example, $v_1, v_2$ interact at $t_1, t_5$. (Right) Each node has its node features (e.g., $\mathbf{x}_1^{\text{node}}$ for $v_1$) and each temporal link has its link features (e.g., $\mathbf{x}_{1,2}^{\text{link}}(t_1), \mathbf{x}_{1,2}^{\text{link}}(t_5)$ are link features between $v_1, v_2$ at $t_1, t_5$). For scenarios without node or link features, we use all-zero vectors instead.

against baselines that are equipped with the RNNs and SAM. In practice, it achieves state-of-the-art performance in terms of different evaluation metrics (e.g., average precision, AUC, Recall@K, and MRR) on real-world temporal graph datasets, with the even smaller number of model parameters and hyper-parameters, and a conceptually simpler input structure and model architecture (Section 4).

Q2: What are the key factors that lead to the success of GraphMixer? We identify three key factors that contribute to the success of GraphMixer: ① *The simplicity of GraphMixer's input data and neural architecture.* Different from most deep learning methods that focus on designing conceptually complicated data preparation techniques and technically complicated neural architectures, we choose to simplifying the neural architecture and utilize a conceptually simpler data as input. Both of which could lead to a better model performance and better generalization (Section 4.4). ② *A time-encoding function that encodes any timestamp as an easily distinguishable input vector for GraphMixer.* Different from most of the existing methods that propose to learn the time-encoding function from the raw input data, our time-encoding function utilizes conceptually simple features and is fixed during training. Interestingly, we show that our fixed time-encoding function is more preferred than the trainable version (used by most previous studies), and could lead to a smoother optimization landscape, a faster convergence speed, and a better generalization (Section 4.2); ③ *A link-encoder that could better distinguish temporal sequences.* Different from most existing methods that summarize sequences using SAM, our encoder module is entirely based on MLPs. Interestingly, our encoder can distinguish temporal sequences that cannot be distinguished by SAM, and it could generalize better due to its simpler neural architecture and lower model complexity (Section 4.3).

To this end, we summarize our contributions as follows: ① We propose a conceptually and technically simple architecture GraphMixer; ② Even without RNN and SAM, GraphMixer not only outperforms all baselines but also enjoys a faster convergence and better generalization ability; ③ Extensive study identifies three factors that contribute to the success of GraphMixer. ④ Our results could motivate future research to rethink the importance of the conceptually and technically simpler method.

## 2    PRELIMINARY AND EXISTING WORKS

**Preliminary.** Figure 1 is an illustration on the temporal graph. Our goal is to predict whether two nodes are connected at a specific timestamp $t_0$ based on all the available temporal graph information happened before that timestamp. For example, to predict whether $v_1, v_2$ are connected at $t_0$, we only have access to the graph structure, node features, and link features with timestamps from $t_1$ to $t_6$.

**Related works.** Most of the temporal graph learning methods are conceptually and technically complicated with advanced neural architectures. It is non-trivial to fully understand the algorithm details without looking into their implementations. Therefore, we select the four most representative and most closely-related methods to introduce and compare them in more details.

- ***JODIE*** Kumar et al. (2019) is a *RNN-based method*. Let us denote $\mathbf{x}_i(t)$ as the embedding of node $v_i$ at time $t$, $\mathbf{x}_{ij}^{\text{link}}(t)$ as the link feature between $v_i, v_j$ at time $t$, and $m_i$ as the timestamp that $v_i$ latest interact with other node. JODIE pre-processes and updates the representation of each node via RNNs (is it just one RNN or multiple RNNs). More specifically, when an interaction between $v_i, v_j$ happens at time $t$, JODIE updates the temporal embedding using RNN by $\mathbf{x}_i(t) = \text{RNN}\big(\mathbf{x}_i(m_i), \mathbf{x}_j(m_j), \mathbf{x}_{ij}^{\text{link}}(t), t - m_i\big)$. Then, the dynamic embedding of node $v_i$ at time $t_0$ is computed by $\mathbf{h}_i(t_0) = (1 + (t_0 - m_i)\mathbf{w}) \cdot \mathbf{x}_i(m_i)$. Finally, the prediction on any node pair at time $t_0$ is computed by $\text{MLP}([\mathbf{h}_i(t_0) \,||\, \mathbf{h}_j(t_0)])$, where $[\cdot||\cdot]$ is the concatenate operation and $\text{MLP}(\mathbf{x})$ is applying 2-layer MLP on $\mathbf{x}$.

- **DySAT** Sankar et al. (2020) is a *SAM-based method*. DySAT requires pre-processing the temporal graph into multiple snapshot graphs by first splitting all timestamps into multiple time-slots, then merging all edges in each time-slot. Let $\mathcal{G}_t(\mathcal{V}, \mathcal{E}_t)$ denote the $t$-th snapshot graph. To capture spatial information, DySAT first applies Graph Attention Network (GAT) Veličković et al. (2018) on each snapshot graph $\mathcal{G}_t$ independently by $\mathbf{X}(t) = \mathtt{GAT}(\mathcal{G}_t)$. Then, to capture of temporal information for each node, Transformer is applied to $\mathbf{x}_i(t) = [\mathbf{X}(t)]_i$ at different timestamps to capture the temporal information by $\mathbf{h}_i(t_k), \ldots \mathbf{h}_i(t_0) = \mathtt{Transformer}(\mathbf{x}_i(t_k), \ldots, \mathbf{x}_i(t_0))$. Finally, the prediction on any node pair at time $t_0$ is computed by $\mathtt{MLP}([\mathbf{h}_i(t_0) \,||\, \mathbf{h}_j(t_0)])$.
- **TGAT** Xu et al. (2020) is a *SAM-based method* that could capture the spatial and temporal information simultaneously. TGAT first generates the time augmented feature of node $i$ at time $t$ by concatenating the raw feature $\mathbf{x}_i$ with a trainable time encoding $\mathbf{z}(t)$ of time $t$, i.e., $\mathbf{x}_i(t) = [\mathbf{x}_i \,||\, \mathbf{z}(t)]$ and $\mathbf{z}(t) = \cos(t\mathbf{w} + \mathbf{b})$. Then, SAM is applied to the time augmented features and produces node representation $\mathbf{h}_i(t_0) = \mathtt{SAM}(\mathbf{x}_i(t_0), \{\mathbf{x}_u(h_u) \mid u \in \mathcal{N}_{t_0}(i)\})$, where $\mathcal{N}_{t_0}(i)$ denotes the neighbors of node $i$ at time $t_0$ and $h_u$ denotes the timestamp of the latest interaction of node $u$. Finally, the prediction on any node pair at time $t_0$ is computed by $\mathtt{MLP}([\mathbf{h}_i(t_0) \,||\, \mathbf{h}_j(t_0)])$.
- **TGN** Rossi et al. (2020) is a mixture of *RNN- and SAM-based method*. In practice, TGN first captures the temporal information using RNN (similarly to JODIE), and then applies graph attention convolution to capture the spatial and temporal information jointly (similarly to TGAT).

Besides, we also consider the following temporal graph learning methods as baselines. These methods could be thought of as an extension on top of the above four most representative methods, but with the underlying idea behind the model design much more conceptually complicated. **CAWs** Wang et al. (2020) is a mixer of *RNN- and SAM- based method* that proposes to represent network dynamics by extracting temporal network motifs using temporal random walks. CAWs replaces node identities with the hitting counts of the nodes based on a set of sampled walks to establish the correlation between motifs. Then, the extracted motifs are fed into RNNs to encode each walk as a representation, and use SAM to aggregate the representations of multi-walks into a single vector for downstream tasks. **TGSRec** Fan et al. (2021) is a *SAM-based method* that proposes to unify sequential patterns and temporal collaborative signals to improve the quality of recommendation. To achieve this goal, they propose to advance the SAM by adopting novel collaborative attention, such that SAM can simultaneously capture collaborative signals from both users and items, as well as consider temporal dynamics inside sequential patterns. **APAN** Wang et al. (2021b) is a *RNN-based method* that proposes to decouple model inference and graph computation to alleviate the damage of the heavy graph query operation to the speed of model inference. More related works are deferred to Appendix B.

## 3 GRAPHMIXER: A CONCEPTUALLY AND TECHNICALLY SIMPLE METHOD

In this section, we first introduce the neural architecture of GraphMixer in Section 3.1 then explicitly highlight its difference to baseline methods in Section 3.2.

### 3.1 DETAILS ON GRAPHMIXER: NEURAL ARCHITECTURE AND INPUT DATA

GraphMixer has three modules: ① *link-encoder* is designed to summarize the information from temporal links (e.g., link timestamps and link features); ② *node-encoder* is designed to summarize the information from nodes (e.g., node features and node identity); ③ *link classifier* predicts whether a link exists based on the output of the aforementioned two encoders.

**Link-encoder.** The link-encoder is designed to summarize the temporal link information associated with each node sorted by timestamps, where temporal link information is referring to the timestamp and features of each link. For example in Figure 1, the temporal link information for node $v_2$ is $\{(t_1, \mathbf{x}_{1,2}^{\text{link}}(t_1)), (t_3, \mathbf{x}_{2,4}^{\text{link}}(t_3)), (t_4, \mathbf{x}_{2,4}^{\text{link}}(t_4)), (t_5, \mathbf{x}_{1,2}^{\text{link}}(t_5))\}$ and for node $v_5$ is $\{(t_2, \mathbf{x}_{3,5}^{\text{link}}(t_2)), (t_6, \mathbf{x}_{4,5}^{\text{link}}(t_6))\}$. In practice, we only keep the top $K$ most recent temporal link information, where $K$ is a dataset dependent hyper-parameter. If multiple links have the same timestamps, we simply keep them the same order as the input raw data. To summarize temporal link information, our link-encoder should have the ability to distinguish different timestamps (achieved by our time-encoding function) and different temporal link information (achieved by the Mixer module).

- *Time-encoding function.* To distinguish different timestamps, we introduce our time-encoding function $\cos(t\boldsymbol{\omega})$, which utilizes features $\boldsymbol{\omega} = \{\alpha^{-(i-1)/\beta}\}_{i=1}^d$ to encode each timestamps into

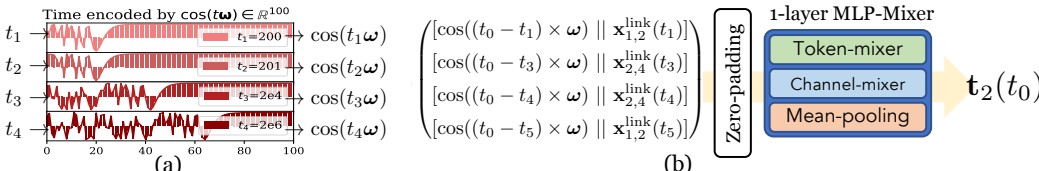

Figure 2: (a) Time-encoding function that pre-process timestamp $t$ into a vector $\cos(t\boldsymbol{\omega})$. The x-axis is the vector dimension and the y-axis is the cosine value. (b) link-encoder takes the temporal link information of node $v_2$ as inputs and outputs a vector $\mathbf{t}_2(t_0)$ that will be used for link prediction.

a $d$-dimensional vector. More specifically, we first map each $t$ to a vector with monotonically exponentially decreasing values $t\boldsymbol{\omega} \in (0, t]$ among the feature dimension, then use cosine function to project all values to $\cos(t\boldsymbol{\omega}) \in [-1, +1]$. The selection of $\alpha, \beta$ is depending on the scale of the maximum timestamp $t_{\max}$ we wish to encode. In order to distinguish all timestamps, we have to make sure $t_{\max} \times \alpha^{-(i-1)/\beta} \to 0$ as $i \to d$ to distinguish all timestamps. In practice, we found $d = 100$ and $\alpha = \beta = \sqrt{d}$ works well for all datasets. Notice that $\boldsymbol{\omega}$ is fixed and will not be updated during training. As shown in Figure 2a, the output of this time-encoding function has two main properties that could help GraphMixer distinguish different timestamps: similar timestamps have similar time-encodings (e.g., the plot of $t_1, t_2$) and the larger the timestamp the later the values in time-encodings converge to $+1$ (e.g., the plot of $t_1, t_3$ or $t_1, t_4$).

- *Mixer for information summarizing.* We use a 1-layer MLP-mixer Tolstikhin et al. (2021) to summarize the temporal link information. Figure 2b is an example on summarizing the temporal link information of node $v_2$. Recall that the temporal link information of node $v_2$ is $\{(t_1, \mathbf{x}_{1,2}^{\text{link}}(t_1)), (t_3, \mathbf{x}_{2,4}^{\text{link}}(t_3)), (t_4, \mathbf{x}_{2,4}^{\text{link}}(t_4)), (t_5, \mathbf{x}_{1,2}^{\text{link}}(t_5))\}$. We first encode timestamps by our time-encoding function then concatenate it with its corresponding link features. For example, we encode $(t_1, \mathbf{x}_{1,2}^{\text{link}}(t_1))$ as $[\cos((t_0 - t_1)\boldsymbol{\omega}) \, || \, \mathbf{x}_{1,2}^{\text{link}}(t_1))]$ where $t_0$ is the timestamp that we want to predict whether the link exists. Then, we stack all the outputs into a big matrix and zero-pad to the fixed length $K$ denoted as $\mathbf{T}_2(t_0)$. Finally, we use an 1-layer MLP-mixer with mean-pooling to compress $\mathbf{T}_2(t_0)$ into a single vector $\mathbf{t}_2(t_0)$. Specifically, the MLP-mixer takes $\mathbf{T}_2(t_0)$ as input

$$\mathbf{T}_2(t_0) \to \mathbf{H}_{\text{input}}, \ \mathbf{H}_{\text{token}} = \mathbf{H}_{\text{input}} + \mathbf{W}_{\text{token}}^{(2)} \texttt{GeLU}(\mathbf{W}_{\text{token}}^{(1)} \texttt{LayerNorm}(\mathbf{H}_{\text{input}})),$$

$$\mathbf{H}_{\text{channel}} = \mathbf{H}_{\text{token}} + \texttt{GeLU}(\texttt{LayerNorm}(\mathbf{H}_{\text{token}})\mathbf{W}_{\text{channel}}^{(1)})\mathbf{W}_{\text{channel}}^{(2)},$$

and output the temporal encoding $\mathbf{t}_2(t_0) = \texttt{Mean}(\mathbf{H}_{\text{channel}})$. Please notice that zero-padding operator is important to capture how often a node interacts with other nodes. The node with more zero-padded dimensions has less temporal linked neighbors. This information is very important in practice according to our experimental observation.

**Node-encoder.** The node-encoder is designed to capture the node identity and node feature information via neighbor mean-pooling. Let us define the 1-hop neighbor of node $v_i$ with link timestamps from $t$ to $t_0$ as $\mathcal{N}(v_i; t, t_0)$. For example in Figure 1, we have $\mathcal{N}(v_2; t_4, t_0) = \{v_1, v_4\}$ and $\mathcal{N}(v_5; t_4, t_0) = \{v_3\}$. Then, the node-info feature is computed based on the 1-hop neighbor by $\mathbf{s}_i(t_0) = \mathbf{x}_i^{\text{node}} + \texttt{Mean}\{\mathbf{x}_j^{\text{node}} \mid v_j \in \mathcal{N}(v_i; t_0 - T, t_0)\}$, where $T$ is a dataset-dependent hyper-parameter. In practice, we found 1-hop neighbors are enough to achieve good performance, and we use one-hot node representations for datasets without node features.

**Link classifier.** Link classifier is designed to classify whether a link exists at time $t_0$ using the output of link-encoder $\mathbf{t}_i(t_0)$ and the output of node-encoder $\mathbf{s}_i(t_0)$. Let us denote the node $v_i$'s representation at time $t_0$ as the concatenation of the above two encodings $\mathbf{h}_i(t_0) = [\mathbf{s}_i(t_0) \, || \, \mathbf{t}_i(t_0)]$. Then, the prediction on whether an interaction between node $v_i, v_j$ happens at time $t_0$ is computed by applying a 2-layer MLP model on $[\mathbf{h}_i(t_0) \, || \, \mathbf{h}_j(t_0)]$, i.e., $p_{ij} = \texttt{MLP}([\mathbf{h}_i(t_0) \, || \, \mathbf{h}_j(t_0)])$.

## 3.2 COMPARISON TO EXISTING METHODS

In the following, we highlight some differences between GraphMixer and other methods, which will be explicitly ablation studied in the experiment section (Section 4.4).

**Temporal graph as undirected graph.** Most of the existing works consider temporal graphs as directed graphs with information only flows from the source node (e.g., users in the recommender system) to the destination nodes (e.g., ads in the recommender system). However, we consider the

Table 1: Comparison on the average precision score for link prediction. GraphMixer uses one-hot node encoding for datasets without node features (marked by ♮). For each dataset, we indicate whether we have the corresponding feature ("L" link features, "N" node features, and "T" link timestamps). Red is the best score, Blue is the best score excluding GraphMixer and its variants.

| | Reddit L, T | Wiki L, T | MOOC T | LastFM T | GDELT L, N, T | GDELT-ne T | GDELT-e N, T |
|---|---|---|---|---|---|---|---|
| JODIE | $99.30 \pm 0.01$ | $98.81 \pm 0.01$ | $99.16 \pm 0.01$ | $67.51 \pm 0.87$ | $98.27 \pm 0.02$ | $97.13 \pm 0.02$ | $96.96 \pm 0.02$ |
| DySAT | $98.52 \pm 0.01$ | $96.71 \pm 0.02$ | $98.82 \pm 0.02$ | $76.40 \pm 0.77$ | $98.52 \pm 0.02$ | $82.47 \pm 0.13$ | $97.25 \pm 0.02$ |
| TGAT | $99.66 \pm 0.01$ | $97.75 \pm 0.02$ | $98.43 \pm 0.01$ | $54.77 \pm 1.01$ | $98.25 \pm 0.02$ | $84.30 \pm 0.10$ | $96.96 \pm 0.02$ |
| TGN | $99.80 \pm 0.01$ | $99.55 \pm 0.01$ | $99.62 \pm 0.01$ | $82.23 \pm 0.50$ | $98.15 \pm 0.02$ | $97.13 \pm 0.02$ | $96.04 \pm 0.02$ |
| CAWs-mean | $98.43 \pm 0.02$ | $97.72 \pm 0.03$ | $62.99 \pm 0.87$ | $76.35 \pm 0.08$ | $95.11 \pm 0.12$ | $69.20 \pm 0.10$ | $91.72 \pm 0.19$ |
| CAWs-attn | $98.51 \pm 0.02$ | $97.95 \pm 0.03$ | $63.07 \pm 0.82$ | $76.31 \pm 0.10$ | $95.06 \pm 0.11$ | $69.54 \pm 0.19$ | $91.54 \pm 0.22$ |
| TGSRec | $95.21 \pm 0.08$ | $91.64 \pm 0.12$ | $83.62 \pm 0.34$ | $76.91 \pm 0.87$ | $97.03 \pm 0.61$ | $97.03 \pm 0.61$ | $97.03 \pm 0.61$ |
| APAN | $99.24 \pm 0.02$ | $98.14 \pm 0.01$ | $98.70 \pm 0.98$ | $69.39 \pm 0.81$ | $95.96 \pm 0.10$ | $97.38 \pm 0.23$ | $96.77 \pm 0.18$ |
| GraphMixer-L | $99.84 \pm 0.01$ | $99.70 \pm 0.01$ | $99.81 \pm 0.01$ | $95.50 \pm 0.03$ | $98.99 \pm 0.02$ | $96.14 \pm 0.02$ | $98.99 \pm 0.02$ |
| GraphMixer-N | $99.24 \pm 0.01$♮ | $90.33 \pm 0.01$♮ | $97.35 \pm 0.02$♮ | $63.80 \pm 0.03$♮ | $94.44 \pm 0.02$ | $96.00 \pm 0.02$♮ | $98.81 \pm 0.02$♮ |
| GraphMixer | $99.93 \pm 0.01$♮ | $99.85 \pm 0.01$♮ | $99.91 \pm 0.01$♮ | $96.31 \pm 0.02$♮ | $98.89 \pm 0.02$ | $98.39 \pm 0.02$♮ | $98.22 \pm 0.02$♮ |

temporal graph as an undirected graph. By doing so, if two nodes are frequently connected in the last few timestamps, the "most recent 1-hop neighbors" sampled for the two nodes on the "undirected" temporal graph would be similar. In other words, the similarity between the sampled neighbors provides information on whether two nodes are frequently connected in the last few timestamps, which is essential for temporal graph link prediction. Intuitively, if two nodes are frequently connected in the last few timestamps, they are also likely to be connected in the recent future.

**Selection on neighbors.** Existing methods consider either "multi-hop recent neighbors" or "multi-hop uniform sampled neighbors", whereas we only consider the "1-hop most recent neighbors". For example, TGAT Xu et al. (2020), DySAT Sankar et al. (2020), and TGSRe Fan et al. (2021) consider multi-hop uniform sampled neighbors; JODIE Kumar et al. (2019), TGN Rossi et al. (2020), and APAN Wang et al. (2021b) maintain the historical node interactions via RNN, which can be think of as multi-hop recent neighbors; CAWs Wang et al. (2020) samples neighbors by random walks, which can also be think of as multi-hop recent neighbors. Although sampling more neighbors could provide a sufficient amount of information for models to reason about, it could also carry much spurious or noisy information. As a result, more complicated model architectures (e.g., RNN or SAM) are required to extract useful information from the raw data, which could lead to a poor model trainability and potentially weaker generalization ability. Instead, we only take the "most recent 1-hop neighbors" into consideration, which is conceptually simpler and enjoys better performance.

## 4 EXPERIMENTS

**Dataset.** We conduct experiments on five real-world datasets, including the *Reddit*, *Wiki*, *MOOC*, *LastFM* datasets that are used in Kumar et al. (2019) and the *GDELT* dataset[1] which is introduced in Zhou et al. (2022). Besides, since *GDELT* is the only dataset with both node and link features, we create its two variants to understand the effect of training data on model performance: *GDELT-e* removes the link feature from *GDELT* and keep the node feature and link timestamps, *GDELT-ne* removes both the link and edge features from *GDELT* and only keep the link timestamps. For each dataset, we use the same $70\%/15\%/15\%$ chronological splits for the train/validation/test sets as existing works. The detailed dataset statistics are summarized in Appendix A.2.

**Baselines.** We compare baselines that are introduced in Section 2. Besides, we create two variants to better understand how node- and link-information contribute to our results, where GraphMixer-L is only using link-encoder and GraphMixer-N is only using node-encoder. We conduct experiments under the transductive learning setting and use average precision for evaluation. The detailed model configuration, training and evaluation process are summarized in Appendix A.3. Due to the space limit, more experiment results on using *Recall@K, MRR, and AUC* as the evaluation metrics, comparison on *wall-clock time* and *number of parameters* are deferred to Appendix C

**Outline.** We first compare GraphMixer with baselines in Section 4.1 then highlight the three key factors that contribute to the success of GraphMixer in Section 4.2, Section 4.3, and Section 4.4.

---

[1]The *GDELT* dataset used in our paper is a sub-sampled version because the original dataset is too big to fit into memory for single-machine training. In practice, we keep 1 temporal link per 100 continuous temporal link.

## 4.1 MAIN EMPIRICAL RESULTS.

**GraphMixer achieves outstanding performance.** We compare the average precision score with baselines in Table 1. We have the following observations: ① GraphMixer outperforms all baselines on all datasets. The experiment results provide sufficient support on our argument that neither RNN nor SAM is necessary for temporal graph link prediction. ② According to the performance of GraphMixer-L on datasets only have link timestamp information (*MOOC*, *LastFM*, and *GDELT-ne*), we know that our time-encoding function could successfully pre-process each timestamp into a meaningful vector. In fact, we will show later in Section 4.2 that our time-encoding function is more preferred than baselines' trainable version. ③ By comparing the performance GraphMixer-N and GraphMixer on *Wiki*, *MOOC*, and *LastFM* datasets, we know that node-encoder alone is not enough to achieve a good performance. However, it provides useful information that could benefit the link-encoder. ④ By comparing the performance of GraphMixer-N on *GDELT* and *GDELT-ne*, we observe that using one-hot encoding outperforms using node features. This also shows the importance of node identity information because one-hot encoding only captures such information. ⑤ More complicated methods (e.g., CAWs, TGSRec, and DDGCL) do not perform well when using the default hyper-parameters[2], which is understandable because these methods have more components with an excessive amount of hyper-parameters to tune.

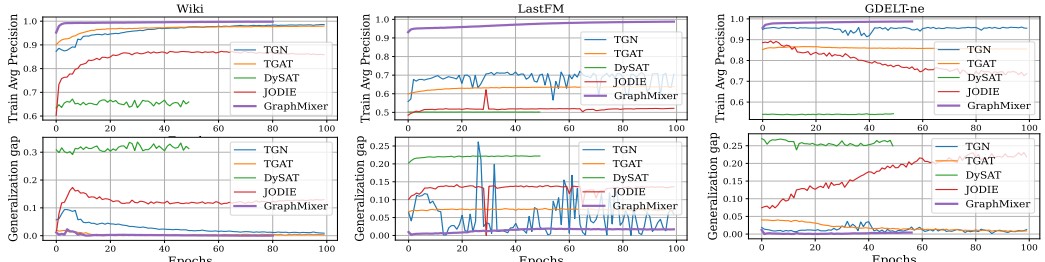

Figure 3: Comparison on the training set average precision and generalization gap for the first 100 training epochs. Results on other datasets can be found in Figure 8.

**GraphMixer enjoys better convergence and generalization ability.** To better understand the model performance, we take a closer look at the dynamic of training accuracy and the generalization gap (the absolute difference between training and evaluation score). The results are reported in Figure 3 and Figure 8: ① The slope of training curves reflects the expressive power and convergence speed of an algorithm. From the first row figures, we can observe that GraphMixer always converge to a high average precision score in just a few epochs, and the training curve is very smooth when compared to baselines. Interestingly, we can observe that the baseline methods cannot always fit the training data, and their training curves fluctuate a lot throughout the training process. ② The generalization gap reflects how well the model could generalize and how stable the model could perform on unseen data (the smaller the better). From the second row figures, the generalization gap curve of GraphMixer is lesser and smoother than baselines, which indicates the generalization power of GraphMixer.

**GraphMixer enjoys a smoother loss landscape.** To understand why *"GraphMixer converges faster and generalizes better, while baselines suffer training unstable issue and generalize poorly"*, we explore the loss landscape by using the visualization tools introduced in Li et al. (2018a). We illustrate the loss landscape in Figure 4 by calculating and visualizing the loss surface along two random directions near the pre-trained optimal parameters. The x- and y-axis indicate how much the optimal solution is stretched along the two random directions, and the optimal point is when x- and y-axis are zero. ① From Figure 4a, 4d, we know GraphMixer enjoys a smoother landscape with a flatter surface at the optimal point, the slope becomes steeper when stretching along the two random directions. The steeper slope on the periphery explains why GraphMixer could converge fast, the flatter surface at the optimal point explains why it could generalize well. ② Surprisingly, we find that baselines have a non-smooth landscape with many spikes on its surface from Figure 4b, 4c, 4e, 4f. This observation provides sufficient explanation on the training instability and poor generalization issue of baselines as shown in Figure 3, 8. Interestingly, as we will show later in Section 4.2, the trainable time-encoding function in baselines is the key to this non-smooth landscape issue. Replacing it with our fixed time-encoding function could flatten the landscape and boost their model performance.

---

[2]In fact, we tried different hyper-parameters based on their default values, but the results are similar.

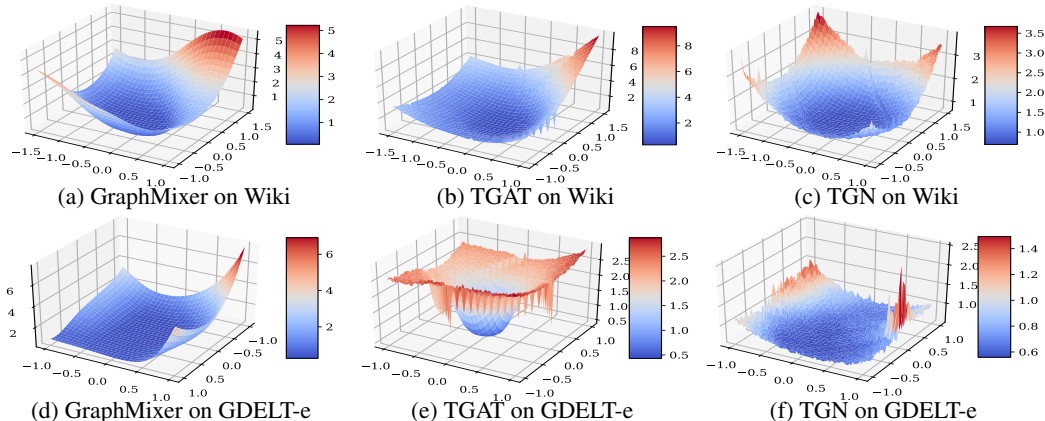

(a) GraphMixer on Wiki     (b) TGAT on Wiki     (c) TGN on Wiki

(d) GraphMixer on GDELT-e     (e) TGAT on GDELT-e     (f) TGN on GDELT-e

Figure 4: Comparison on the training loss landscape. Results on other datasets and other baselines can be found in Appendix E.

## 4.2 ON THE IMPORTANCE OF OUR FIXED TIME-ENCODING FUNCTION.

Existing works (i.e., JODIE, TGAT, and TGN) leverage a trainable time-encoding function $\mathbf{z}(t) = \cos(t\mathbf{w}^\top + \mathbf{b})$ to represent timestamps[3]. However, we argue that using trainable time-encoding function could cause instability during training because its gradient $\frac{\partial \cos(t\mathbf{w}+\mathbf{b})}{\partial \mathbf{w}} = t \times \sin(t\mathbf{w} + \mathbf{b})$ scales proportional to the timestamps, which could lead to training instability issue and cause the baselines' the non-smooth landscape issue as shown in Figure 4. As an alternative, we utilize the fixed time-encoding function $\mathbf{z}(t) = \cos(t\boldsymbol{\omega})$ with fixed features $\boldsymbol{\omega}$ that could capture the relative difference between two timestamps (introduced in Section 3.1). To verify this, we introduce a simple experiment to test whether the time-encoding functions (both our fixed version and baselines' trainable version) are expressive enough, such that a simple linear classifier can distinguish the time-encodings of two different timestamps produced by the time-encoding functions. Specially, our goal is to classify if $t_1 > t_2$ by learning a linear classifier on $[\mathbf{z}(t_1) \,||\, \mathbf{z}(t_2)]$. During training, we randomly generate two timestamps $t_1, t_2 \in [0, 10^6]$ and ask a fully connected layer to classify whether a timestamp is greater than another. As shown in Figure 5a, using the trainable time-encoding function (orange curve) will suffer from the unstable exploding gradient issue (left upper figure) and its performance remains almost the same during the training process (left lower figure). However, using our fixed time-encoding function (blue curve) does not have the unstable exploding gradient issue and can quickly achieve high accuracy within several iterations. Meanwhile, we compare the parameter trajectories of the two models in Figure 5b. We observe that the change of parameters on the trainable time-encoding function is drastically larger than our fixed version. A huge change in weight parameters could deteriorate the model's performance. Most importantly, by replacing baselines' trainable time-encoding function with our fixed version, most baselines have a smoother optimization landscape (Figure 6) and a better model performance (in Table 2), which further verifies our argument that our fixed time-encoding function is more preferred than the trainable version.

Table 2: Comparison on average precision score with fixed/trainable time encoding function (TEF). The results before "→" is for trainable TEF (same as Table 1) and after "→" is for fixed TEF.

|  | Reddit | Wiki | MOOC | LastFM | GDELT-ne | GDELT-e |
|---|---|---|---|---|---|---|
| JODIE | $99.30 \to \mathbf{99.76}$ | $98.81 \to \mathbf{99.00}$ | $99.16 \to \mathbf{99.17}$ | $67.51 \to \mathbf{79.89}$ | $97.13 \to \mathbf{98.23}$ | $96.96 \to \mathbf{96.96}$ |
| TGAT | $98.66 \to \mathbf{99.48}$ | $96.71 \to \mathbf{98.55}$ | $98.43 \to \mathbf{99.33}$ | $54.77 \to \mathbf{76.26}$ | $84.30 \to \mathbf{92.31}$ | $\mathbf{96.96} \to 96.28$ |
| TGN | $99.80 \to \mathbf{99.83}$ | $\mathbf{99.55} \to 99.54$ | $99.62 \to \mathbf{99.62}$ | $82.23 \to \mathbf{87.58}$ | $98.15 \to \mathbf{98.25}$ | $96.04 \to \mathbf{97.34}$ |

## 4.3 ON THE IMPORTANCE OF MLP-MIXER IN GRAPHMIXER'S LINK-ENCODER

In this section, we aim to achieve a deeper understanding on the expressive power of the link-encoder by answering the following two questions: "*Can we replace the MLP-mixer in link-encoder with self-attention?*" and "*Why MLP-mixer is a good alternative of self-attention?*" To answer these questions,

---

[3]In fact, other baselines (e.g., CAWs, TGSRec, APAN) also utilize this trainable time-encoding function. However, we focus our discussion on the selected methods for the ease of presentation.

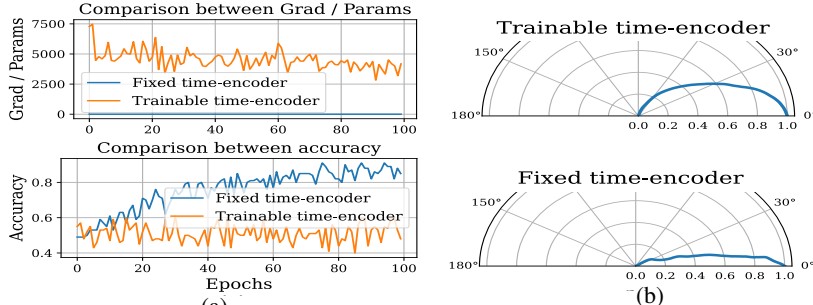

(a)

(b)

Figure 5: (a) Comparison on the *gradient / parameters norm* and *accuracy* at each iteration. (b) Comparison on the trajectories of parameter change, where the radius is $r_t = \|\boldsymbol{\delta}_t\|/\|\boldsymbol{\delta}_0\|$, the angle is $\theta_t = \arccos\langle\boldsymbol{\delta}_t/\|\boldsymbol{\delta}_t\|_2, \boldsymbol{\delta}_0/\|\boldsymbol{\delta}_0\|_2\rangle$, and $\boldsymbol{\delta}_t = \mathbf{w}_t - \mathbf{w}^\star$ is the difference between $\mathbf{w}_t$ to optimal point $\mathbf{w}^\star$. The more the model parameters change during training, the larger the semicircle.

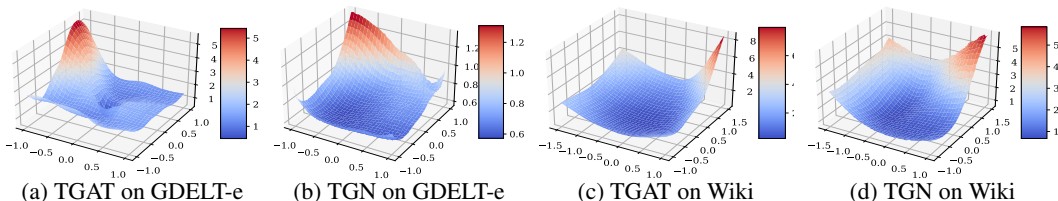

(a) TGAT on GDELT-e     (b) TGN on GDELT-e     (c) TGAT on Wiki     (d) TGN on Wiki

Figure 6: Comparison on the training loss landscape *fixed time-encoding function*. Results on other datasets and baselines can be found in Appendix F.

let us first conduct experiments by replacing the MLP-mixer in link-encoder with full/1-hop self-attention and sum/mean-pooling, where full self-attention is widely used in Transformers and 1-hop self-attention is widely used in graph attention networks. As shown in Table 3, GraphMixer suffers from performance degradation when using self-attention: the best performance is achieved when using MLP-mixer with zero-padding, while the model performance drop slightly when using self-attention with sum-pooling (row 2 and 4), and the performance drop significantly when using self-attention with mean-pooling (row 3 and 5). Self-attention with mean-pooling has a weaker model performance because it cannot distinguish "temporal sequences with identical link timestamps and features" (e.g., cannot distinguish $[a_1, a_1]$ and $[a_1]$ and it cannot explicitly capture "the length of temporal sequences" (e.g., cannot distinguish if $[a_1, a_2]$ is longer than $[a_3]$), which are both very important for GraphMixer understand how frequent a node interacts with other nodes. We explicitly verify this in Figure 7 by first generating two temporal sequences (with timestamps but without link features), then encoding the timestamps into vectors via time-encoding function, and asking full self-attention and MLP-mixer to distinguish. As shown in Figure 7, self-attention with mean-pooling cannot distinguish two temporal sequences with identical timestamps (because all the self-attention weights are equivalent if the features of the node on the two sides of a link are identical) and cannot capture the sequence length (because of mean-pooling simply averages the inputs and does not take the input size into consideration). However, MLP-mixer in GraphMixer can distinguish the above two sequences because of zero-padding. Fortunately, the aforementioned two weaknesses could be alleviated by replacing the mean-pooling in temporal self-attention with the sum-pooling, which explains why using sum-pooling brings better model performance than mean-pooling. However, since self-attention modules have more parameters and are harder to train, they could generalize poor when the downstream task is not too complicated.

## 4.4 KEY FACTORS TO THE BETTER PERFORMANCE

One of the major factors that contributes to GraphMixer's success is the simplicity of GraphMixer's neural architecture and input data. Using conceptually simple input data that better aligned with their labels allows a simple neural network model to capture the underlying mapping between the input to their labels, which could lead to a better generalization ability. In the following, we explicitly verify this by comparing the performance of GraphMixer with different input data in Table 4: ① Recall from Section 3.2 that the "most recent 1-hop neighbors" sampled for the two nodes on the "undirected" temporal graph could provide information on whether two nodes are frequently connected in the last few timestamps, which is essential for temporal graph link prediction. To verify this, we conduct

Table 3: Comparison on the average precision score. ♮ use 20 neighbors due to out of GPU memory.

| Link-info encoder with | | Reddit | Wiki | MOOC | LastFM | GDELT-ne |
|---|---|---|---|---|---|---|
| (Default) MLP-mixer | + Zero-padding | **99.93** | **99.85** | **99.91** | **96.31** | **98.39** |
| Full self-attention | + Sum pooling | 99.81 | 98.19 | 99.55 | 93.97 | 98.28♮ |
| | + Mean pooling | 99.00 | 98.05 | 99.31 | 89.15 | 97.13♮ |
| 1-hop self-attention | + Sum pooling | 99.81 | 98.01 | 99.30 | 93.69 | 98.16 |
| | + Mean pooling | 98.94 | 97.29 | 98.96 | 72.32 | 97.09 |

(a) e.g., classify if $[a_1, a_1] = [a_1]$  (b) e.g., classify if $\text{size}[a_1, a_2] > \text{size}[a_3]$

Figure 7: (a) We generate identical timestamp sequences with different length, then ask MLP-mixer and GAT to distinguish whether the generated sequence are identical (b) We generate random sequence with different length, then ask MLP-mixer and GAT to classify which sequence is longer.

ablation study by comparing the model performance on direct and undirected temporal graphs. As shown in the 1st and 2nd row of Table 4, changing from undirected to direct graph results in a significant performance drop because such information is missing. ② Recall from Section 3.1 that instead of feeding the raw timestamp to GraphMixer and encoding each timestamp with a trainable time-encoding function, GraphMixer encodes the timestamps via our fixed time-encoding function and feed the encoded representation to GraphMixer, which reduces the model complexity of learning a time-encoding function from data. This could be verified by the 3rd and 4th rows of Table 4, where using the pre-encoded time information could give us a better performance. ③ Selecting the input data that has similar distribution in training and evaluation set could also potentially improve the evaluation error. For example, using relative timestamps (i.e., each neighbor's timestamp is subtracted by its root node's timestamp) is better than absolute timestamps (e.g., using Unix timestamp)because the absolute timestamps in the evaluation set and training set are from different range when using chronological splits, but they are very likely to overlap if using relative timestamps. As shown in the 3rd to 6th rows of Table 4, using relative time information always gives a better model performance than using absolute time information. ④ Selecting the most representative neighbors for each node. For example, we found 1-hop most recent interacted neighbors are the most representative for link prediction. Switching to either 2-hop neighbors or uniform sampled neighbors will hurt the model performance according to the 7th to the 10th row of Table 4.

Table 4: Comparison on the average precision score of GraphMixer with different input data. The highlighted rows are identical to our default setting.

| | | | Reddit | Wiki | MOOC | LastFM |
|---|---|---|---|---|---|---|
| Direct vs undirect temporal graph | | Directed temporal graph | 99.69 | 88.37 | 97.87 | 78.34 |
| | | Undirected temporal graph | **99.93** | **99.85** | **99.91** | **96.31** |
| Time information | | Relative timestamp $(t_i - t_0)$ | 99.79 | 99.80 | 99.81 | 95.32 |
| | | Relative time-encoding $\cos((t_i - t_0)\boldsymbol{\omega})$ | **99.93** | **99.85** | **99.91** | **96.31** |
| | | Absolute timestamp $t_i$ | 98.90 | 98.23 | 98.73 | 92.25 |
| | | Absolute time-encoding $\cos(t_i\boldsymbol{\omega})$ | 99.52 | 99.13 | 99.74 | 95.28 |
| Neighbor selection | 2-hop | Most recent neighbors | 99.39 | 98.05 | 99.11 | 89.36 |
| | 1-hop | Most recent neighbors | **99.93** | **99.85** | **99.91** | **96.31** |
| | 2-hop | Uniform sample neighbors | 97.66 | 92.57 | 98.87 | 65.72 |
| | 1-hop | Uniform sample neighbors | 98.19 | 94.74 | 98.40 | 60.02 |

## 5 CONCLUSION

In this paper, we propose a conceptually and technically simple architecture GraphMixer for temporal link prediction. GraphMixer not only outperforms all baselines but also enjoys a faster convergence speed and better generalization ability. An extensive study identifies three key factors that contribute to the success of GraphMixer and highlights the importance of simpler neural architecture and input data structure. An interesting future direction, not limited to temporal graph learning, is designing algorithms that could automatically select the best input data and data pre-processing strategies for different downstream tasks.

## ACKNOWLEDGEMENTS

This work was supported in part by NSF grant 2008398. Majority of this work was completed during Weilin Cong's internship at Meta AI under the mentorship of Si Zhang. We also extend our gratitude to Long Jin for his co-mentorship and for his contribution to the idea of using MLP-Mixer on graphs.

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

## A  EXPERIMENT SETUP DETAILS

### A.1  HARDWARE SPECIFICATION AND ENVIRONMENT

We run our experiments on a single machine with Intel i9-10850K, Nvidia RTX 3090 GPU, and 64GB RAM memory. The code is written in Python 3.8 and we use PyTorch 1.12.1 on CUDA 11.6 to train the model on the GPU. Implementation details could be found at

https://github.com/CongWeilin/GraphMixer.

### A.2  DETAILS ON DATASET

The dataset used in this paper could be automatically downloaded by this script. Reddit dataset[4] consists of one month of posts made by users on subreddits. The link feature is extracted by converting the text of each post into a feature vector. Wikipedia dataset[5] consists of one month of edits made by edits on Wikipedia pages. The link feature is extracted by converting the edit test into an LIWC-feature vector. LastFM dataset[6]: consists of one month of who listens-to-which song information. MOOC dataset[7] consists of actions done by students on a MOOC online course. GDELT dataset[8] is a temporal knowledge graph dataset originated from the Event Database which records events happening in the world from news and articles.

Table 5: Dataset statistic.

|  | $|\mathcal{V}|$ | $|\mathcal{E}|$ | $(t_{\max} - t_{\min})/|\mathcal{E}|$ | $\dim(\mathbf{x}_i^{\mathrm{node}})$ | $\dim(\mathbf{x}_{ij}^{\mathrm{link}})$ | Node features | Link features | Timestamps |
|---|---|---|---|---|---|---|---|---|
| Reddit | 10,984 | 672,447 | 4 | 0 | 172 | No | Yes | Yes |
| Wiki | 9,227 | 157,474 | 17 | 0 | 172 | No | Yes | Yes |
| MOOC | 7,144 | 411,749 | 3.6 | 0 | 0 | No | No | Yes |
| LastFM | 1,980 | 1,293,103 | 106 | 0 | 0 | No | No | Yes |
| GDELT | 8,831 | 1,912,909 | 0.1 | 413 | 186 | Yes | Yes | Yes |
| GDELT-ne | 8,831 | 1,912,909 | 0.1 | 0 | 0 | No | No | Yes |
| GDELT-n | 8,831 | 1,912,909 | 0.1 | 0 | 186 | No | Yes | Yes |

### A.3  MODEL CONFIGURATIONS, TRAINING AND EVALUATION PROCESS

**Baseline implementations.** The implementation on JODIE, DySAT, TGAT, TGN, and APPN follows the temporal graph learning framework Zhou et al. (2022)[9]. Compared to the original baselines' implementation, this framework's implementation could achieve a better overall score than its original implementation. The implementation of CAWs-mean and CAWs-attn follows their official implementation[10], we choose the number of random walk steps from $8, 16, 32$ to balance the training time. The implementation of TGSRec follows their official implementation[11]. The implementation of DDGCL follows their official implementation[12]. We directly test using their official implementation by changing our data structure to their required structure and using their default hyper-parameters.

**GraphMixer implementation.** We implement GraphMixer under the TGL framework Zhou et al. (2022) and use their default hyper-parameters (e.g., learning rate $0.0001$, weight decay $10^{-6}$, batch size 600, hidden dimension 100, etc) to achieve a fair comparison. In GraphMixer, there are only two hyper-parameters as introduced in Section 3: The number of 1-hop most recent neighbors $K$ and the time-slot size $T$. In practice, hyper-parameter $T$ is set the time-gap of the last $2,000$ interactions, which is fixed for all datasets; hyper-parameter $K = 10$ for Reddit and LastFM, $K = 20$ for MOOC, and $K = 30$ for GDELT and Wiki.

---

[4]Download from http://snap.stanford.edu/jodie/reddit.csv

[5]Download from http://snap.stanford.edu/jodie/wikipedia.csv

[6]Download from http://snap.stanford.edu/jodie/lastfm.csv

[7]Download from http://snap.stanford.edu/jodie/mooc.csv

[8]Download from https://github.com/amazon-research/tgl/blob/main/down.sh

[9]The TGL framework can be found at https://github.com/amazon-research/tgl

[10]CAW's official implementation can be found at https://github.com/snap-stanford/CAW

[11]TGSRec's official implementation can be found at https://github.com/DyGRec/TGSRec

[12]DDGCL's official implementation can be found at https://github.com/ckldan520/DDGCL

**Training and evaluation.** A unified training and evaluation process (e.g., mini-batch and data preparation) is used for GraphMixer and baselines. Specifically, each mini-batch is constructed by first sampling a set of positive node pairs and an equal amount of negative node pairs. Then, an algorithm-dependent node sampler is used to sample the neighboring of each mini-batch node and computed their node representation based on the sampled neighborhood. Finally, we concatenate each node pair and use the link prediction classifier (introduced in Section 3.1) for binary classification. We conduct experiments under the transduction learning setting and use average precision for evaluation.

## B MORE DISCUSSION ON EXISTING TEMPORAL GRAPH METHODS

### B.1 RECENT METHODS THAT WE DO NOT COMPARE WITH

There are other temporal graph learning algorithms that are related to the temporal link prediction task but we did not compare GraphMixer with them because (1) the official implementation of some of the above works are not released by the authors and we could not reproduce their results as reported in the paper, and (2) we already compare many recent baselines that we believe it is enough to verify the success of GraphMixer.

For example, *MeTA* Wang et al. (2021c) proposes data augmentation to overcome the over-fitting issue in temporal graph learning. More specifically, they generate a few graphs with different data augmentation magnitudes and perform the message passing between these graphs to provide adaptively augmented inputs for every prediction. *TCL* Wang et al. (2021a) proposes to use a transformer to separately extract the temporal neighborhoods representations associated with the two interaction nodes and then utilizes a co-attentional transformer to model inter-dependencies at a semantic level. To boost model performance, contrastive learning is used to maximize mutual information between the predictive representations of two future interaction nodes. *TNS* Wang et al. (2021d) proposes a temporal-aware neighbor sampling strategy that can provide an adaptive receptive neighborhood for every node at any time. *LSTSR* Chi et al. (2022) propose Long Short-Term Preference Modeling for Continuous-Time Sequential Recommendation to capture the evolution of short-term preference under dynamic graph. *DyRep* Trivedi et al. (2019) uses RNNs to propagate messages in interactions to update node representations. *DynAERNN* Goyal et al. (2018) uses a fully connected layer to first encode the network representation, then pass the encoded features to the RNN, and use the fully connected network to decode the future network structure. *VRGNN* Hajiramezanali et al. (2019) generalizes variational GAE Kipf & Welling (2016) to temporal graphs, which makes priors dependent on historical dynamics and captures these dynamics using RNN. *EvolveGCN* Pareja et al. (2020) uses RNN to estimate GCN parameters for future snapshots. *DDGCL* Tian et al. (2021) is a *SAM-based method* that propose a debiased GAN-type contrastive loss as the learning objective to correct the sampling bias that occurred in the negative sample construction process of temporal graph learning. *NAT* Luo & Li (2022) maintain two sets of representations for each node, i.e., node representations and link representations. For each node, NAT not only preserve a node representation, but also keep node pair representations for a subset of neighbors of the node. *PINT* Souza et al. (2022) using 1-WL test and proposes temporal encoding to boost the expressive power.

### B.2 WHY EXISTING METHODS ARE CONCEPTUALLY AND TECHNICALLY COMPLICATED?

Please notice that we are not claiming conceptually and technically complicated is bad. Instead, we are simply suggesting that the conceptually and technically complicated simpler methods might be more preferred than the complicated one if they could achieve similar performance.

- We say a method is conceptually complicated if the underlying idea behind the method is non-trivial. For example, *CAWs* represents network dynamics by "motifs extracted using temporal random walks", represents node identity by "hitting counts of the nodes based on a set of sampled walks"; **TGSRec** takes "temporal collaborative signals" into consideration. These concepts are non-trivial to understand in the first place and could potentially require much domain knowledge from other fields to understand the behavior of the method.
- We say a method is technically complicated if the method is non-trivial to implement due to many hyper-parameters and many details that need to be taken care of, which could potentially make the application to a real-world scenario challenge. For example, *JODIE* and *TGN* require maintaining a "memory" for each node by using RNN, and this "memory" needs to be reset every time after

evaluation because then it might carry information about the evaluation data. ***CAWs*** extracts features by using multiple temporal random walks, which makes implementing and hyper-parameter fine-tuning more challenging. ***MeTA*** and ***TCL*** consider many data augmentation strategies, each of which is not trivial to implement and could affect the model's performance in different ways.

### B.3 THEORETICAL WORKS ON TEMPORAL GRAPH LEARNING

Recently, researchers have investigated the expressive power of temporal graph neural networks using graph isomorphism tests. For instance, Gao & Ribeiro (2022) have categorized temporal graph learning methods into "time-and-graph" and "time-then-graph" and compared their expressiveness. They have demonstrated that "time-then-graph" outperforms "time-and-graph" in terms of 1-WL test expressive power. This partially explains why GraphMixer has shown good performance, as it can be thought of as a "time-then-graph" algorithm. Additionally, Souza et al. (2022) have shown that incorporating temporal encoding and using the 1-WL test can enhance the expressive power of temporal graph neural networks.

## C    MORE EXPERIMENT RESULTS

### C.1    COMPARISON ON CONVERGENCE SPEED AND GENERALIZATION

We include the missing figures of Section 4. Results on other datasets and the discussion on the experiment results could be found next to Figure 3.

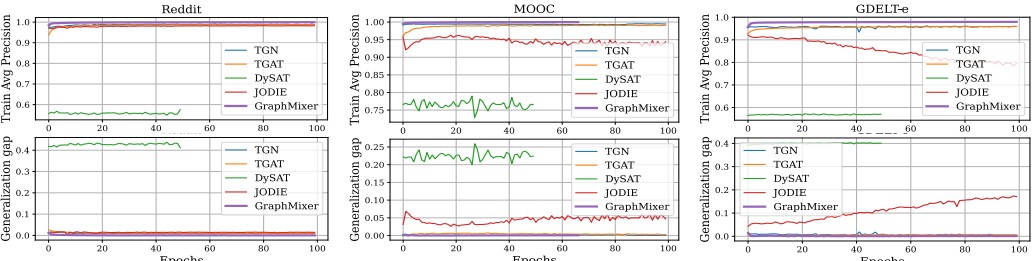

Figure 8: Comparison of the link prediction training average precision and generalization gap for the first 100 training epochs. Results on other datasets can be found in Figure 3.

### C.2    TRANSDUCTIVE LEARNING WITH RECALL@K AND MRR AS EVALUATION METRIC

Recall@K and MRR (mean reciprocal rank) are popular evaluation metrics used in the real-world recommendation system. The larger the numbers, the better the model performance. Our Recall@K and MRR is implemented based on the Open Graph Benchmark's link prediction evaluation metrics' implementation[13]. More specifically, we first sample 100 negative destination nodes for the source node of each temporal link node pair, then our goal is to rank the positive temporal link node pairs higher than 100 negative destination nodes. In the following, we compare the Recall@5 and MRR score of GraphMixer with the selected four most representative baselines. Please notice that since these methods are implemented under the same framework, the model performance is evaluated by using the same model used in Table 1, the comparison is guaranteed to be fair.

Table 6: Comparison on the Recall@K and MRR.

|  | Reddit | | Wiki | | MOOC | | LastFM | | GDELT | |
|---|---|---|---|---|---|---|---|---|---|---|
|  | R@5 | MRR | R@5 | MRR | R@5 | MRR | R@5 | MRR | R@5 | MRR |
| JODIE | 0.9181 | 0.6271 | 0.9098 | 0.7752 | 0.9818 | 0.7551 | 0.2034 | 0.1206 | 0.8554 | 0.6048 |
| DySAT | 0.9189 | 0.7774 | 0.8889 | 0.7561 | 0.9989 | 0.7906 | 0.4159 | 0.3322 | 0.8302 | 0.4236 |
| TGAT | 0.9774 | 0.8709 | 0.8508 | 0.6132 | 0.9736 | 0.7425 | 0.1040 | 0.0769 | 0.3513 | 0.2366 |
| TGN | 0.9787 | 0.9093 | 0.8878 | 0.8016 | 0.9904 | 0.9904 | 0.1649 | 0.1153 | 0.9297 | 0.7295 |
| GraphMixer | **1.0** | **0.9965** | **0.9972** | **0.9876** | **0.9999** | **0.9910** | **0.9998** | **0.9649** | **0.9930** | **0.8934** |

---

[13]https://github.com/snap-stanford/ogb/blob/master/ogb/linkproppred/evaluate.py

We have the following observations on Table 1:

- According to the results in Table 6, GraphMixer could achieve outstanding performance across all datasets. Especially on the LastFM and GDELT datasets (denser graphs than other datasets). This might implies GraphMixer is more suitable for denser graphs than other baseline methods.
- The results of some baseline methods behave less better with Recall@K and MRR evaluation metrics. For example, TGN on LastFM dataset, TGAT on LastFM and GDELT, etc. The above results also imply the limitation of only considering average precision and AUC score for temporal link evaluation.

## C.3 COMPARISON ON WALL-CLOCK TIME

In the following, we compare the wall-clock time it takes for GraphMixer and baselines to finish a single epoch of training.

Table 7: Comparison on the wall-clock computation time for single-epoch of training.

|            | Reddit     | Wiki     | MOOC     | LastFM     | GDELT      |
|------------|-----------|----------|----------|-----------|------------|
| JODIE      | 5 sec     | 2 sec    | 4 sec    | 11 sec    | 16 sec     |
| DySAT      | 33 sec    | 6 sec    | 16 sec   | 41 sec    | 83 sec     |
| TGAT       | 15 sec    | 4 sec    | 8 sec    | 28 sec    | 41 sec     |
| TGN        | 8 sec     | 2 sec    | 5 sec    | 15 sec    | 32 sec     |
| CAWs-mean  | 1,893 sec | 277 sec  | 641 sec  | 1,797 sec | 4,544 sec  |
| CAWs-attn  | 1,930 sec | 282 sec  | 653 sec  | 1,832 sec | 4,634 sec  |
| TGSRec     | 538 sec   | 157 sec  | 656 sec  | 1,810 sec | 3,707 sec  |
| APAN       | 13 sec    | 4 sec    | 9 sec    | 28 sec    | 25 sec     |
| GraphMixer | 12 sec    | 3 sec    | 7 sec    | 21 sec    | 32 sec     |

When comparing the computation time of GraphMixer with CAWs, TGSRec, and DDGCL, we found that GraphMixer takes significantly lesser time than these baselines, which indicates the effectiveness of GraphMixer. When comparing the computation time of GraphMixer with JODIE, DySAT, TGAT, APAN, and TGN, we found that GraphMixer is very close to or even slightly faster than some baseline methods. Our computation time is slightly slower than other baseline methods (e.g., JODIE and TGN) mainly because these baselines are using well-optimized computation functions from DGL Wang et al. (2019), while GraphMixer is just using a composition of basic PyTorch functions. In fact, according to the Table 8, GraphMixer has a similar/smaller amount of parameters with these baselines.

Table 8: Comparison on the number of model parameters.

|                            | GraphMixer | GraphMixer-T | GraphMixer-S | Jodie | DySAT | TGAT | TGN  |
|----------------------------|------------|--------------|--------------|-------|-------|------|------|
| # Parameters ($\times 10^5$) | 2.25       | 1.42         | 2.07         | 1.21  | 9.38  | 3.80 | 3.54 |

Besides, our current implementation also need to preprocess the input data for each epoch of training, e.g., sorting nodes in subgraph according to temporal order and removing duplicated edges. For example, the data preparation at each epoch takes 41 sec on Reddit, 9 sec on Wiki, 20 sec on MOOC, 48 sec on LastFM, and 71 sec on GDELT. However, by caching the pre-processed data in the memory, we only need to pre-process the input data at the first epoch of the training process because our neighbor selection is deterministic and the input data does not change at each epoch.

## C.4 TRANSDUCTIVE LEARNING WITH AUC AS EVALUATION METRIC

AUC (Under the ROC Curve) is one of the most widely accepted evaluation metric for link prediction, which has been used in many existing works Xu et al. (2020); Rossi et al. (2020). In the following, we compare the AUC score of GraphMixer with baselines. We have the following observations: ① GraphMixer outperforms all baselines on all datasets. In particular, GraphMixer attains more than 1% gain over all baselines on the *LastFM*, *GDELT-ne*, and *GDELT-e* datasets, attains around 2% gain over non-RNN methods *DySAT* and *TGAT* on the *Wiki* dataset, and attains around 11% gain over non-RNN methods *DySAT* and *TGAT* on the *GDELT-ne* dataset. The experiment results provide sufficient support on our argument that neither RNN nor SAM is necessary for temporal graph link prediction. ② According to the performance of GraphMixer-L on datasets only have link timestamp information (*MOOC*, *LastFM*, and *GDELT-ne*), we know that our time-encoding function could successfully pre-process each timestamp into a meaningful vector. ③ By comparing the performance GraphMixer-N and GraphMixer on *Wiki*, *MOOC*, and *LastFM* datasets, we know that node-info

encoder alone is not enough to achieve a good performance. However, it provides useful information that could benefit the link-info encoder. ④ By comparing the performance of GraphMixer-N on *GDELT* and *GDELT-ne*, we observe that using one-hot encoding outperforms using node features. This also shows the importance of node identity information because one-hot encoding only captures such information.

Table 9: Comparison on the AUC score for link prediction. GraphMixer uses one-hot node encoding for datasets without node features (marked by ♮). For each dataset we indicate whether we have the corresponding feature ("L" link features, "N" node features, and "T" link timestamps).

| | Reddit
L, T | Wiki
L, T | MOOC
T | LastFM
T | GDELT
L, N, T | GDELT-ne
T | GDELT-e
N, T |
|---|---|---|---|---|---|---|---|
| JODIE | $99.30 \pm 0.01$ | $98.81 \pm 0.01$ | $99.16 \pm 0.01$ | $67.51 \pm 1.99$ | $98.55 \pm 0.01$ | $97.13 \pm 0.02$ | $96.96 \pm 0.02$ |
| DySAT | $98.52 \pm 0.01$ | $96.71 \pm 0.02$ | $98.82 \pm 0.01$ | $76.40 \pm 0.81$ | $98.52 \pm 0.01$ | $82.47 \pm 0.04$ | $97.25 \pm 0.02$ |
| TGAT | $99.66 \pm 0.01$ | $97.75 \pm 0.02$ | $98.43 \pm 0.01$ | $54.77 \pm 1.02$ | $98.25 \pm 0.01$ | $84.30 \pm 0.03$ | $96.96 \pm 0.02$ |
| TGN | $99.80 \pm 0.01$ | $99.55 \pm 0.01$ | $99.62 \pm 0.01$ | $82.23 \pm 0.50$ | $98.15 \pm 0.01$ | $97.13 \pm 0.02$ | $96.04 \pm 0.02$ |
| CAWs-mean | $98.18 \pm 0.01$ | $97.25 \pm 0.03$ | $63.88 \pm 0.92$ | $72.92 \pm 0.33$ | $95.19 \pm 0.09$ | $71.82 \pm 0.08$ | $91.40 \pm 0.19$ |
| CAWs-attn | $98.30 \pm 0.01$ | $97.89 \pm 0.02$ | $63.95 \pm 0.81$ | $72.93 \pm 0.54$ | $95.13 \pm 0.11$ | $71.82 \pm 0.08$ | $91.64 \pm 0.24$ |
| TGSRec | $94.74 \pm 0.20$ | $91.32 \pm 0.19$ | $80.70 \pm 2.31$ | $76.66 \pm 1.54$ | $96.72 \pm 0.42$ | $96.72 \pm 0.42$ | $96.72 \pm 0.42$ |
| APAN | $99.24 \pm 0.01$ | $97.25 \pm 0.01$ | $98.58 \pm 0.01$ | $62.73 \pm 0.64$ | $96.46 \pm 0.11$ | $98.39 \pm 0.17$ | $97.85 \pm 0.19$ |
| GraphMixer-L | $99.84 \pm 0.01$ | $99.70 \pm 0.01$ | $99.87 \pm 0.01$ | $97.04 \pm 0.02$ | $\mathbf{98.99} \pm 0.02$ | $96.54 \pm 0.02$ | $\mathbf{98.99} \pm 0.02$ |
| GraphMixer-N | $99.53 \pm 0.01^♮$ | $91.49 \pm 0.01^♮$ | $98.66 \pm 0.02^♮$ | $71.51 \pm 0.03^♮$ | $94.44 \pm 0.02$ | $96.00 \pm 0.02^♮$ | $98.81 \pm 0.02^♮$ |
| GraphMixer | $\mathbf{99.94} \pm 0.01^♮$ | $\mathbf{99.82} \pm 0.01^♮$ | $\mathbf{99.93} \pm 0.01^♮$ | $\mathbf{97.38} \pm 0.02^♮$ | $98.89 \pm 0.02$ | $\mathbf{98.50} \pm 0.02^♮$ | $98.48 \pm 0.02^♮$ |

### C.5 BASELINES WITH UNDIRECTED TEMPORAL GRAPH

GraphMixer utilizes undirected temporal graph to capture whether two nodes are frequently connected in the last few timestamps. In the following, we test whether using undirected temporal graph could improve the performance of baseline methods. As we can see from Table 10, using undirected temporal graph cannot improve the performance of baseline methods much because such information are already implicitly captured via their neural architecture design or sampling methods.

Table 10: Comparison on baselines with undirected temporal graph (Average precision | AUC score).

| | Reddit | Wiki | MOOC | LastFM |
|---|---|---|---|---|
| JODIE | 99.24 \| 99.40 | 98.91 \| 99.07 | 99.18 \| 99.53 | 73.25 \| 80.24 |
| DySAT | 98.53 \| 98.42 | 96.61 \| 96.88 | 98.80 \| 99.26 | 76.23 \| 73.90 |
| TGAT | 99.70 \| 99.73 | 97.35 \| 97.68 | 98.41 \| 98.85 | 54.58 \| 57.03 |
| TGN | 99.84 \| 99.87 | 99.59 \| 99.61 | 99.44 \| 99.66 | 91.96 \| 93.20 |

## D DISCUSSION ABOUT MODEL PERFORMANCE ON LASTFM

Our results show that GraphMixer could outperform baselines on LastFM dataset with a large margin, which is due to a composite effect of multiple factors. In the following, we summarize several potential factors that lead to our observation on the model performance.

- **Larger average time-gap.** LastFM has a larger average time-gap $(t_{\max} - t_{\min})/|\mathcal{E}|$ than other datasets. As shown in the dataset statistic (Table 5), LastFM has an average time gap of 106, which is significantly larger than other datasets. For example, Reddit's average time gap is 4, Wiki's average time gap is 17, MOOC's average time gap is 3.6, and GDELT's average time gap is 0.1. Since baseline methods are relying on RNN and SAM to process the historical temporal information, they implicitly assumes the temporal information is "smooth" and with smaller average time gap. Therefore, baseline methods could potentially work better on the dataset with a smaller average time gap but are less ideal on LastFM. GraphMixer is not relying on RNN or SAM, therefore could be less affected by the aforementioned issue.
- **Larger average node degree.** LastFM has a larger average node degree $|\mathcal{E}|/|\mathcal{V}|$ than other datasets, which potentially prone to over-smoothing (aggregating features from many neighbors make output representation less distinguishable Li et al. (2018b)), over-squashing (aggregating much information into a limited memory might compress too much useful information Alon & Yahav (2020)) and over-fitting Cong et al. (2021a) effect. For example, according to the dataset statistic in Table 5, LastFM has an average node degree of 653, which is larger than the Reddit's average node degree 61, Wiki's average node degree 17, MOOC's average node degree 57, and GDELT's average node degree 216. Existing methods either use the memory cell in RNN to store temporal information or use SAM to aggregate temporal information from multi-hops,

which could be less ideal on a dense graph due to over-smoothing and over-squashing. However, GraphMixer is less relying on the aggregation schema, therefore its performance is better than the baseline methods.

- **Larger maximum timestamp.** GraphMixer is using fixed time encoder but baselines are using trainable time-encoders. Since the largest timestamp $t_{\max}$ in LastFM is larger than other datasets, the trainable time-encoder is more affected by the unbounded gradient issue as discussed in Table 2 and Section 4.2. For example, the $t_{\max}$ in LastFM is 137 millon, while $t_{\max}$ in GDELT 0.2 millon, $t_{\max}$ in Reddit 2.6 millon, $t_{\max}$ in Wiki 2.6 millon, and $t_{\max}$ in MOOC 2.6 millon.

# E    MISSING FIGURES ON LOSS LANDSCAPE

## E.1    LOSS LANDSCAPE ON WIKI

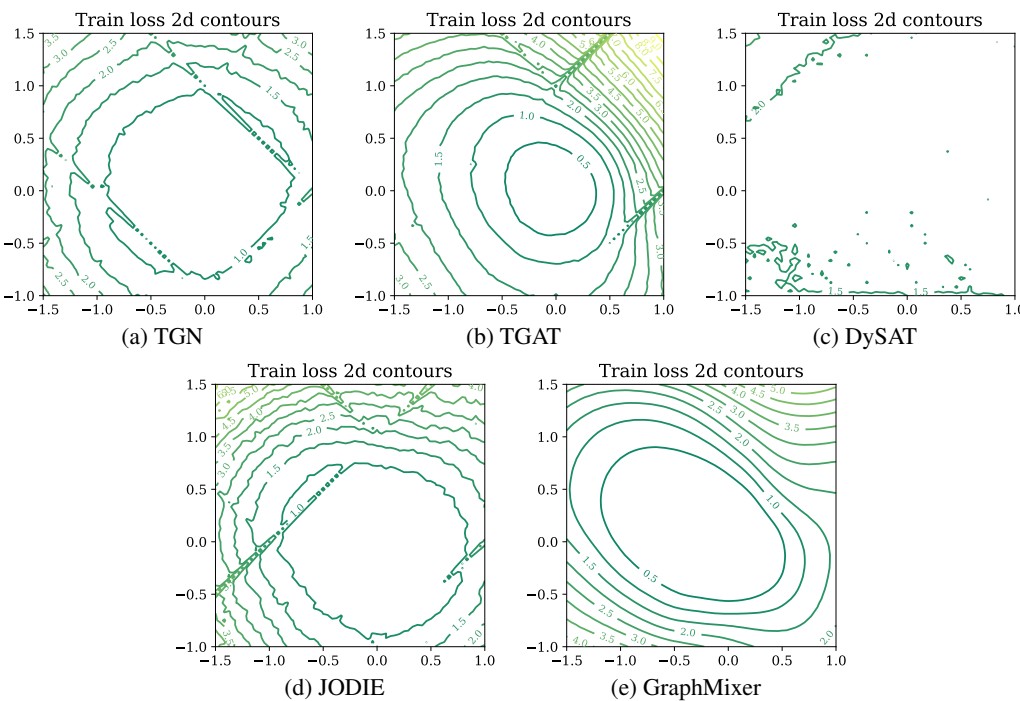

Figure 9: Comparison on the training loss landscape (Contour) on **Wiki** Dataset.

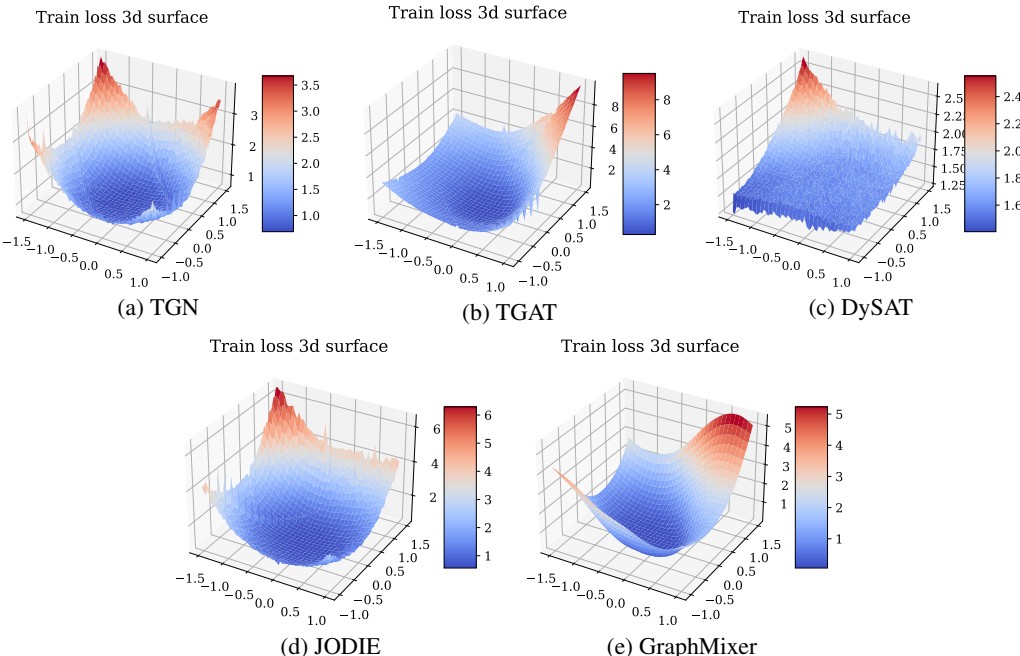

Figure 10: Comparison on the training loss landscape (Surface) on **Wiki** Dataset.

### E.2 Loss landscape on Reddit

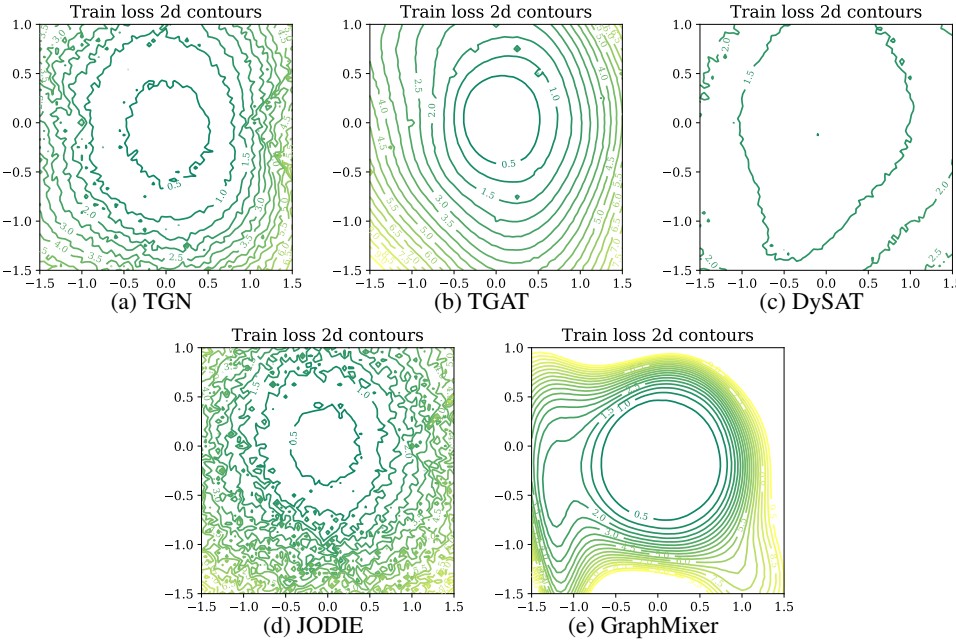

Figure 11: Comparison on the training loss landscape (Contour) on **Reddit** Dataset.

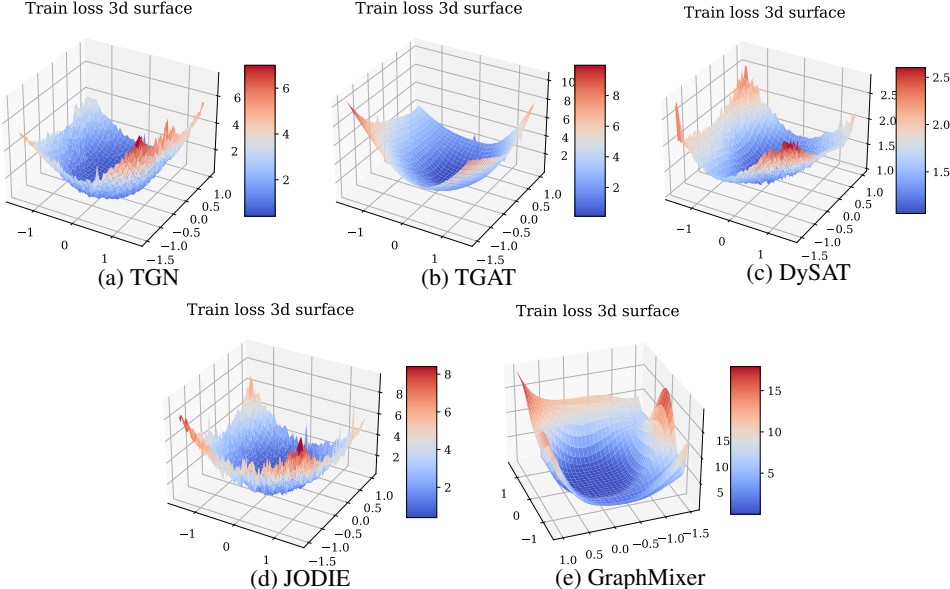

Figure 12: Comparison on the training loss landscape (Surface) on **Reddit** Dataset.

### E.3    LOSS LANDSCAPE ON MOOC

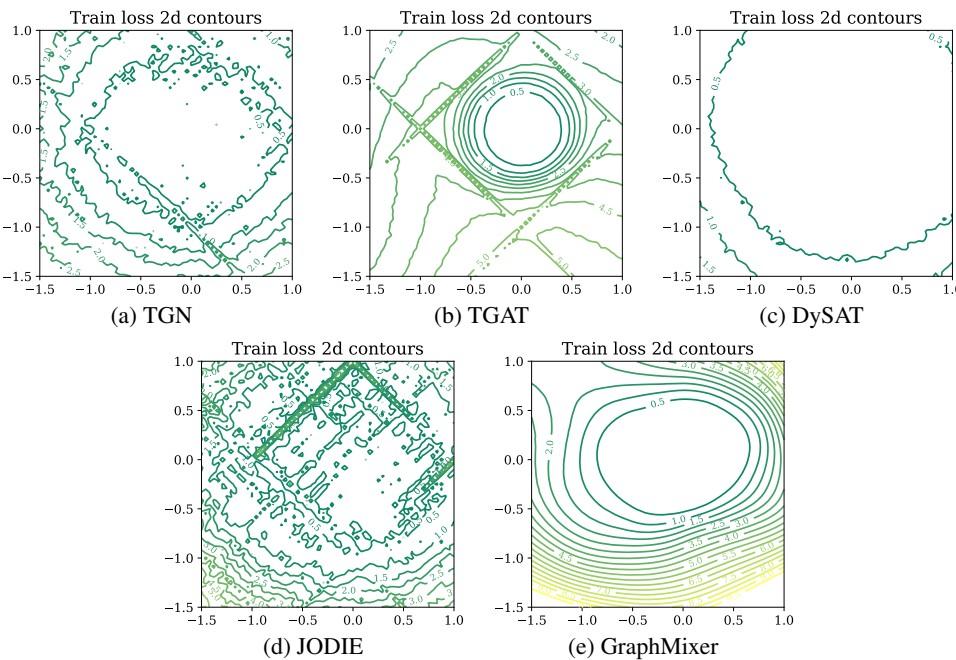

Figure 13: Comparison on the training loss landscape (Contour) on **MOOC** Dataset.

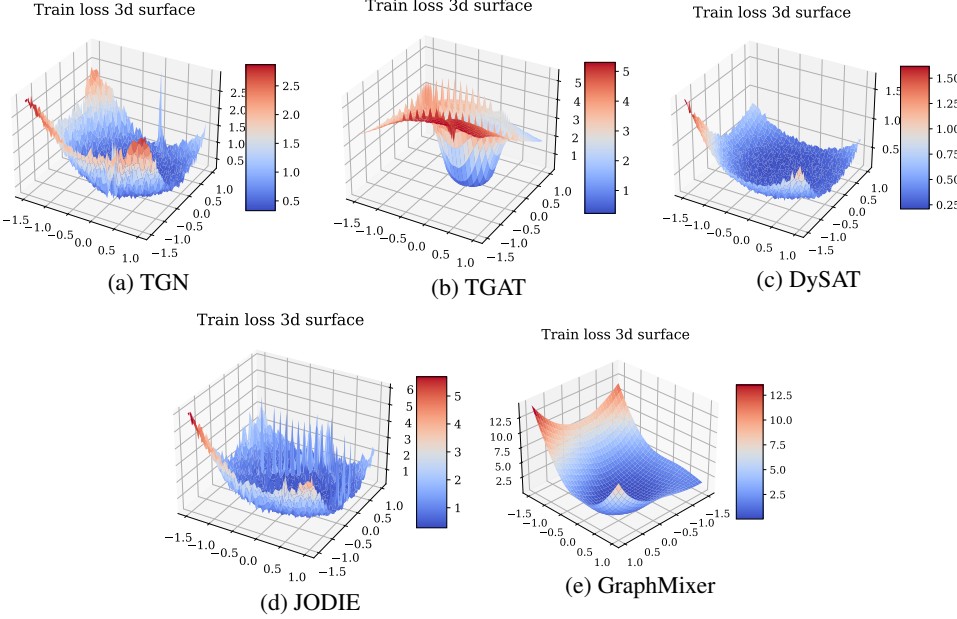

Figure 14: Comparison on the training loss landscape (Surface) on **MOOC** Dataset.

### E.4 LOSS LANDSCAPE ON LASTFM

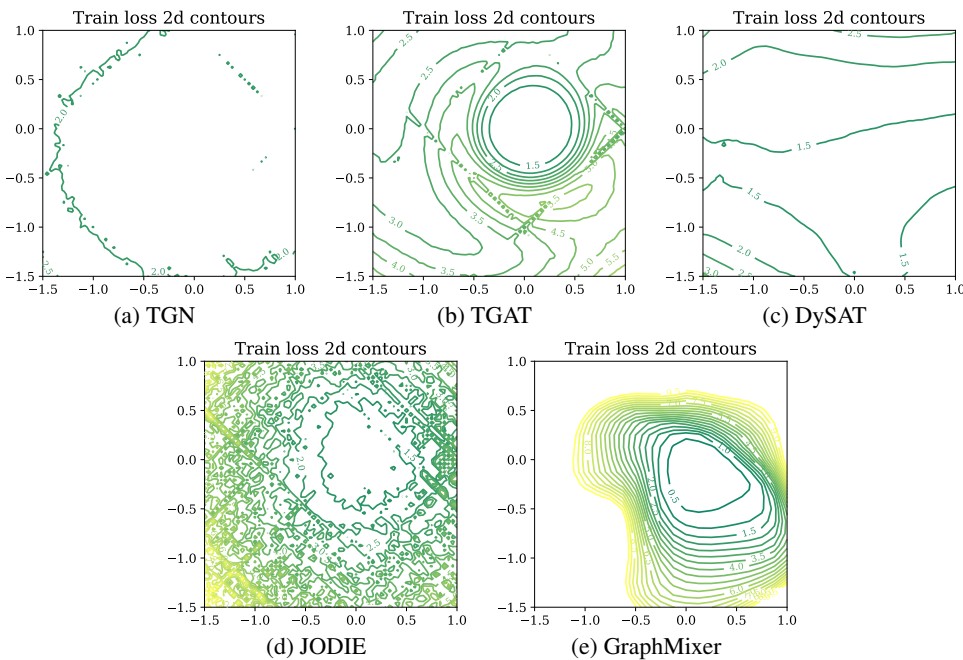

Figure 15: Comparison on the training loss landscape (Contour) on **LastFM** Dataset.

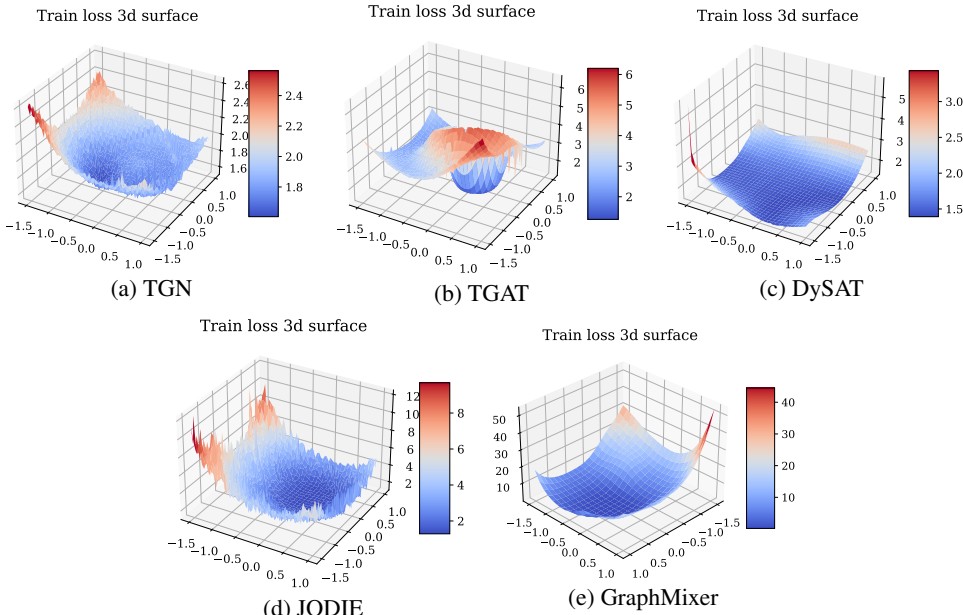

Figure 16: Comparison on the training loss landscape (Surface) on **LastFM** Dataset.

### E.5   LOSS LANDSCAPE ON GDELT-NE

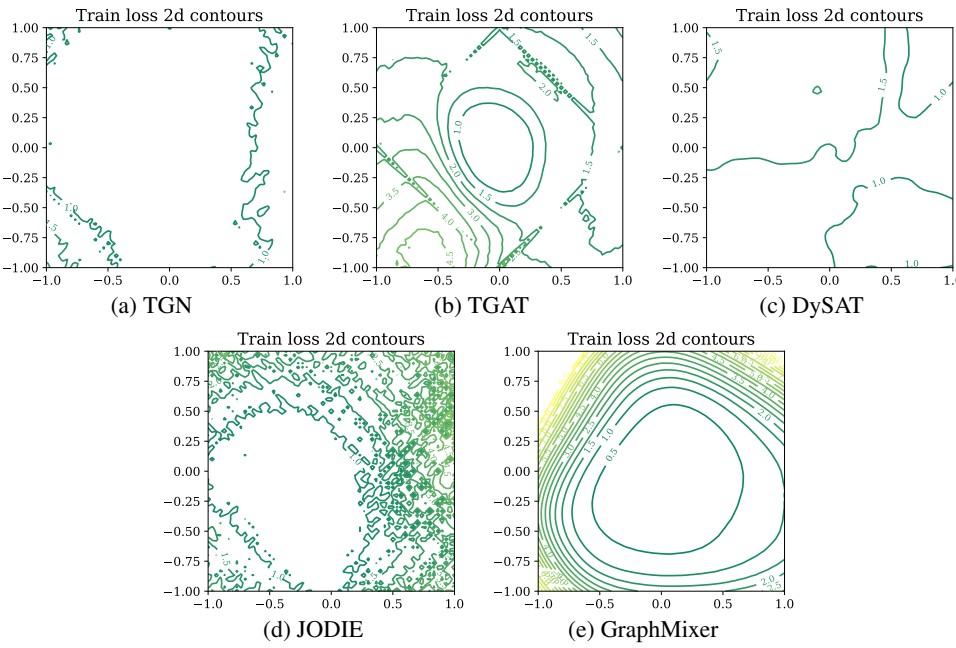

Figure 17: Comparison on the training loss landscape (Contour) on **GDELT-ne** Dataset.

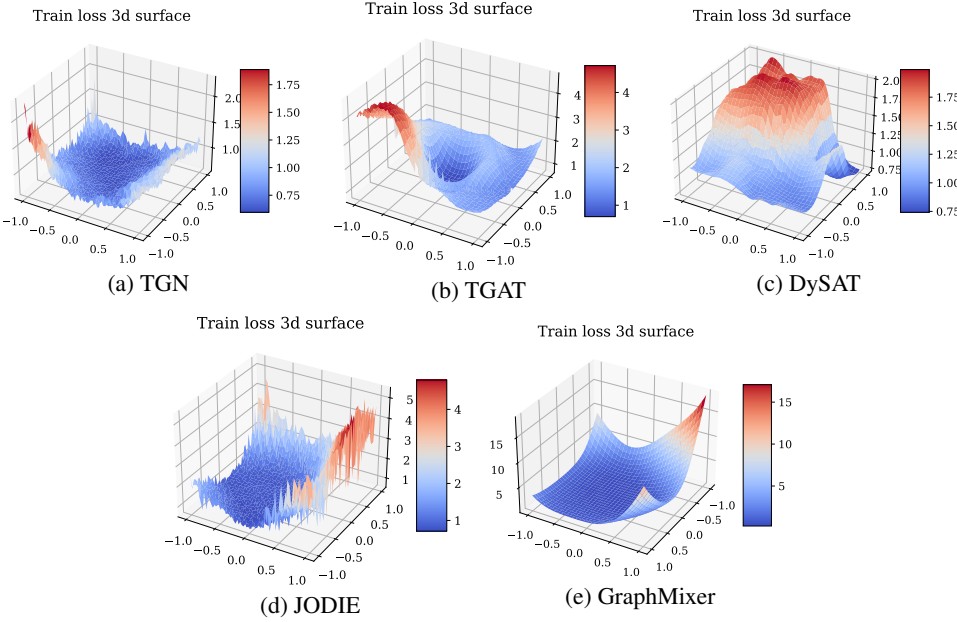

Figure 18: Comparison on the training loss landscape (Surface) on **GDELT-ne** Dataset.

### E.6    LOSS LANDSCAPE ON GDELT-E

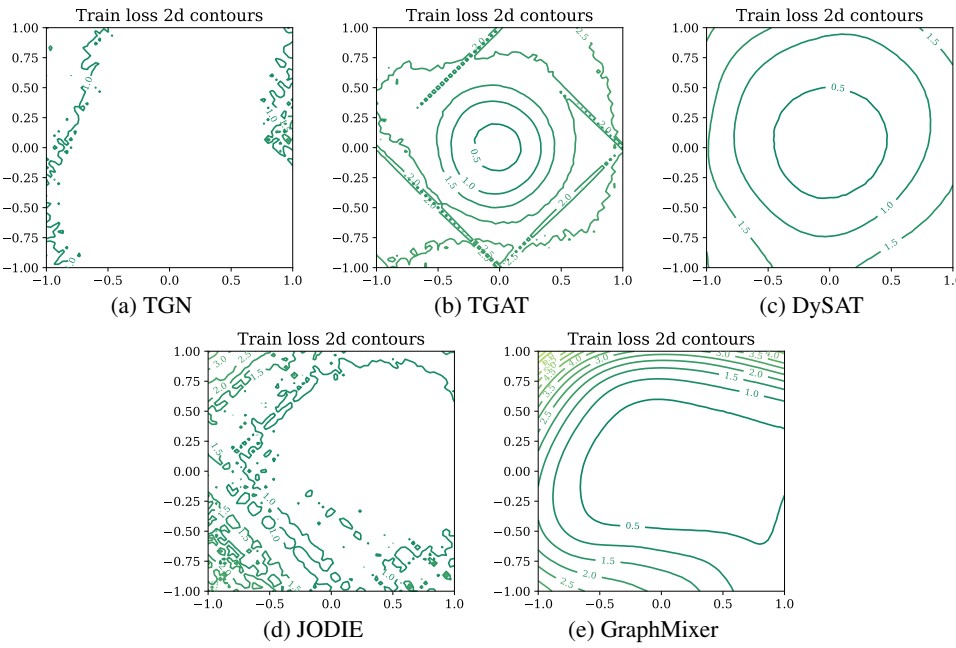

Figure 19: Comparison on the training loss landscape (Contour) on **GDELT-e** Dataset.

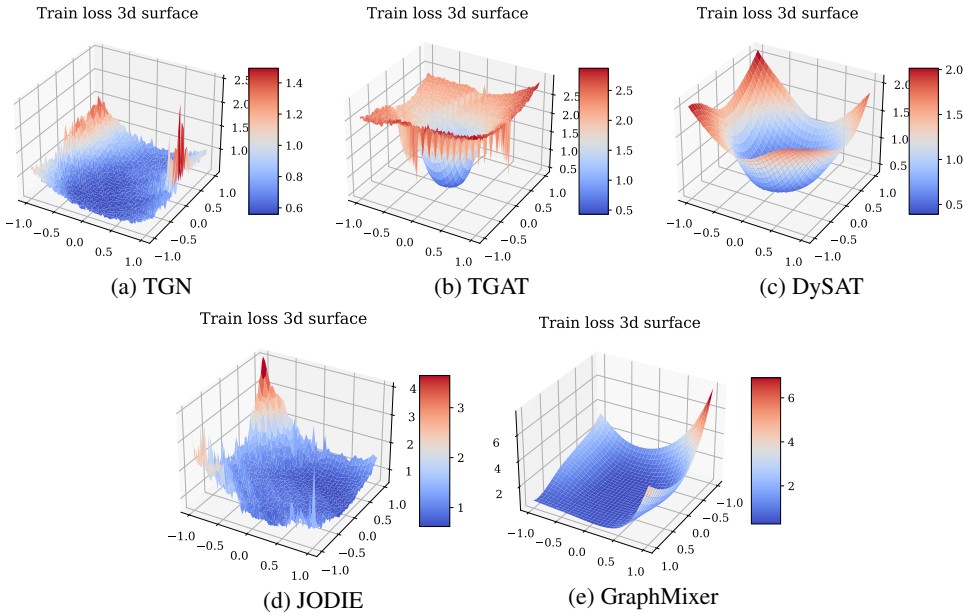

Figure 20: Comparison on the training loss landscape (Surface) on **GDELT-e** Dataset.

## F MISSING FIGURES ON LOSS LANDSCAPE WITH OUR TIME-ENCODING FUNCTION

### F.1 TGAT WITH FIXED TIME-ENCODING FUNCTION

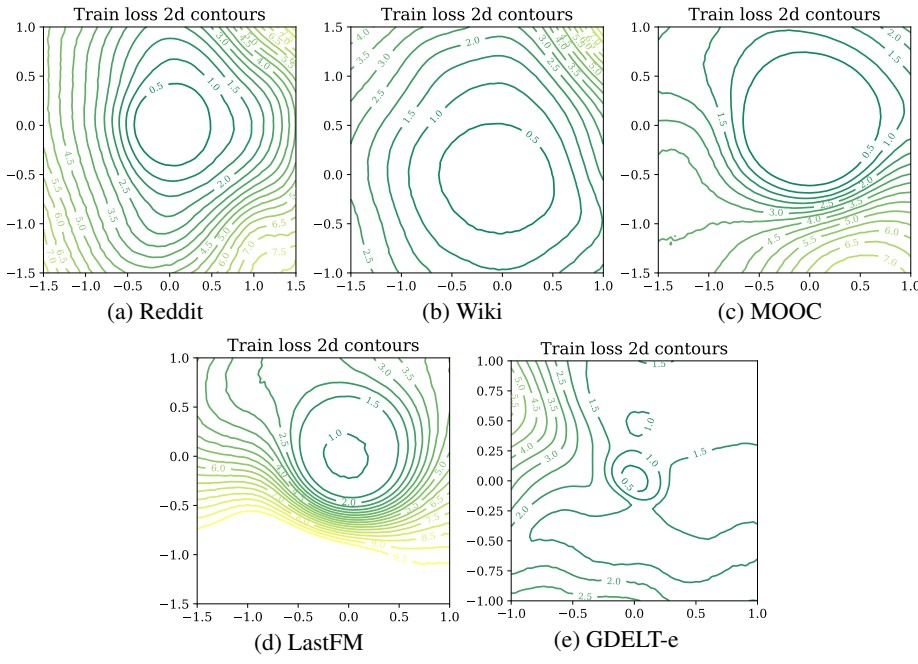

Figure 21: Training loss landscape (Contour) of **TGAT** with *fixed time-encoding function*.

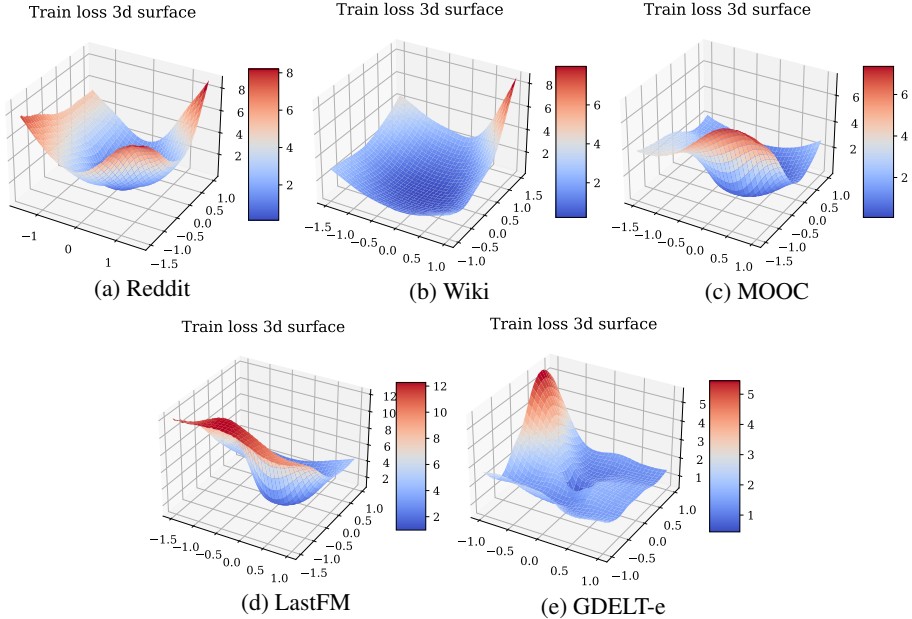

Figure 22: Training loss landscape (Surface) of **TGAT** with *fixed time-encoding function*.

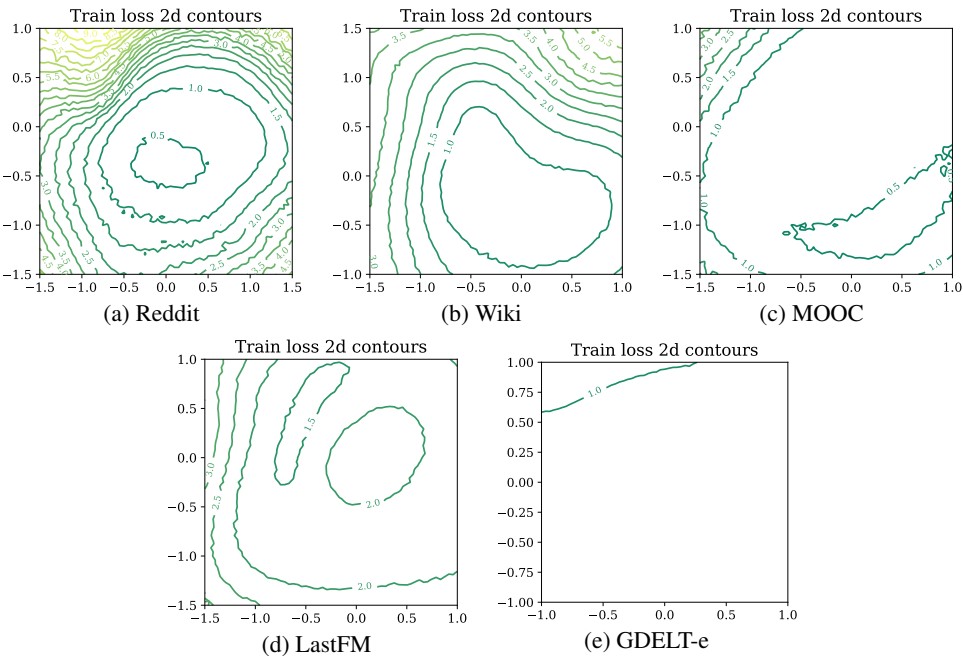

Figure 23: Training loss landscape (Contour) of **TGN** with *fixed time-encoding function*.

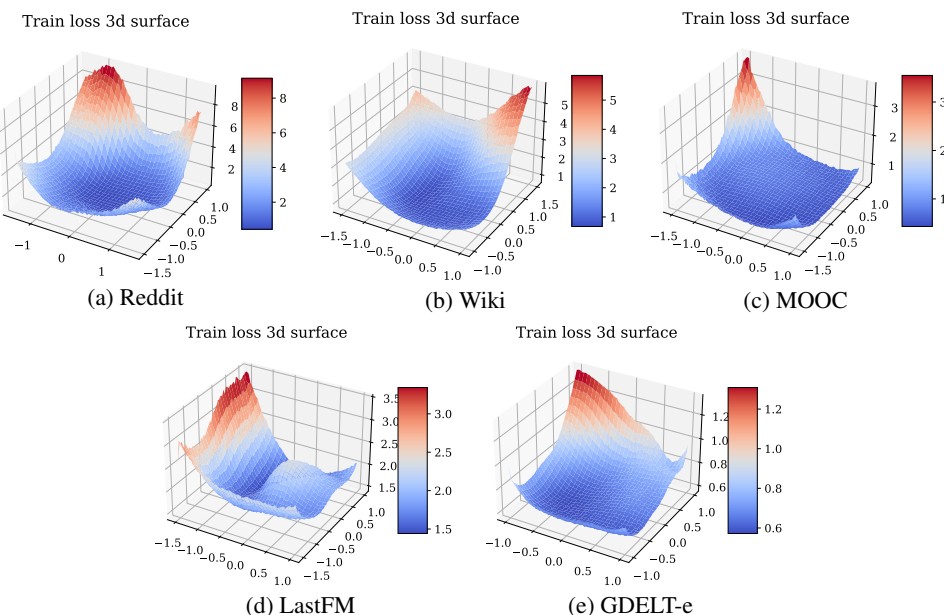

Figure 24: Training loss landscape (Surface) of **TGN** with *fixed time-encoding function*.

### F.3 JODIE WITH FIXED TIME-ENCODING FUNCTION

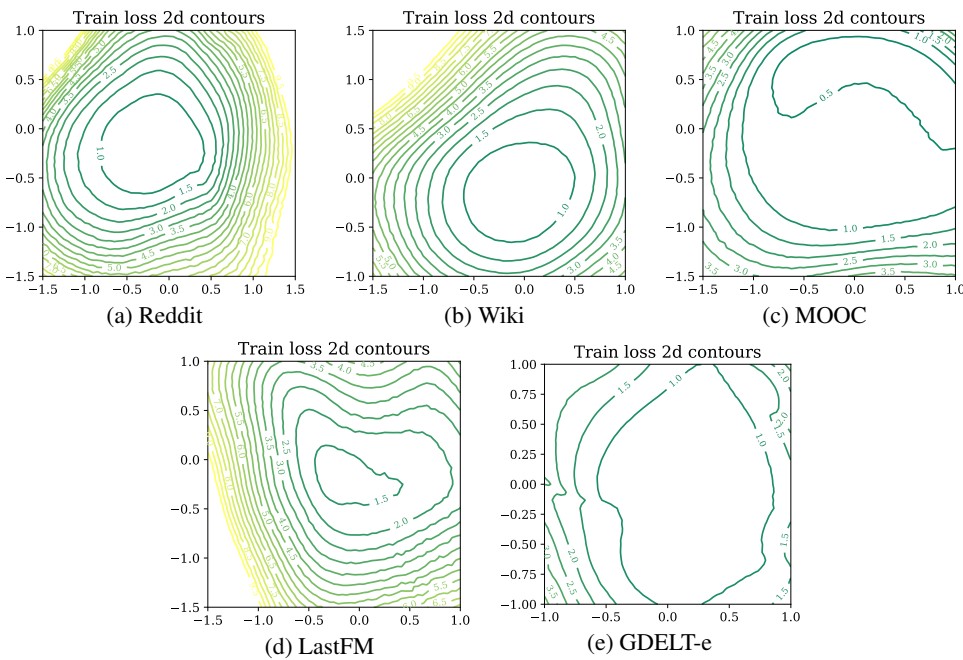

Figure 25: Training loss landscape (Contour) of **JODIE** with *fixed time-encoding function*.

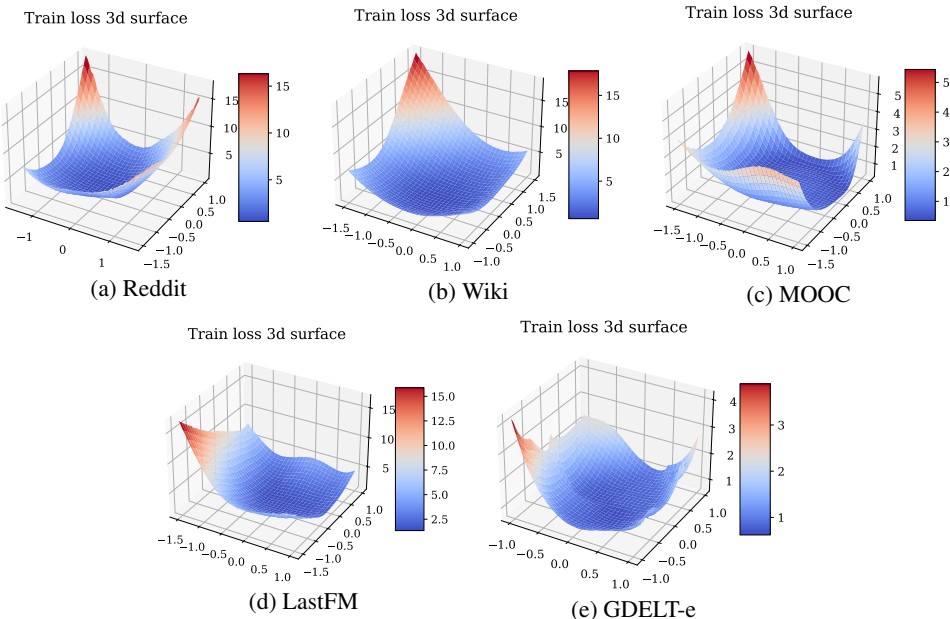

Figure 26: Training loss landscape (Surface) of **JODIE** with *fixed time-encoding function*.

