# OpenReview forum: "Do We Really Need Complicated Model Architectures For Temporal Networks?"
_ICLR.cc/2023/Conference — ICLR 2023 notable top 5%_

### Official Review · Reviewer_7yc7 · 2022-10-23

**Confidence:** 3
**Correctness:** 3
**Technical Novelty And Significance:** 3
**Empirical Novelty And Significance:** 3
**Recommendation:** 8

**Clarity, Quality, Novelty And Reproducibility:**

Though I am not the expert in graph learning area, I enjoyed reading this paper. I would like to give this paper an borderline accept rating. I think a simple but effective method satisfied me.

**Strength And Weaknesses:**

First of all, I'm not the expert in the graph learning area and my research focus is computer vision. From my side, I would to say: A simple method along with state-of-the-art performances satisfies me.

When I read this paper, I'm satisfied by the simple method, the detailed experiments. Thus I would like to give an accept rating.

However, there is still a little bit concerns:

1. I think the major novelty lies in the time-encoding part, also the proposed GraphMixer performs extremely better on LastFM dataset (c.f., Table 2). Could the author give explanations of why GraphMixer work well on LastFM? Since I only observe marginal improvements on other datasets.

**Summary Of The Paper:**

This paper proposes a simple architecture named GraphMixer for temporal graph learning. The GraphMixer is based on simple MLP structure and achieves state-of-the-art performances on temporal link prediction benchmarks.

**Summary Of The Review:**

I think this paper is a simple and effective paper, thus I want to give it an accept rating.

---

> ### Author Response · Authors · 2022-11-10
> **Response to Reviewer 7yc7**
>
> Thank you for being interested in our work.
>
> **Q1. Further discussion on why GraphMixer works a lot better on LastFM than baselines?**
>
> A1: In general, the model performance is a composite effect of multiple factors. In the following, we summarize several potential factors that could make LastFM more challenging for baselines.
> - **Larger average time-gap size.** LastFM has a larger average time-gap $(t_{\max} - t_{\min})/|\mathcal{E}|$ than other datasets. As shown in the dataset statistics (Table 5), LastFM has an average time gap of 106, which is significantly larger than other datasets. For example, Reddit's average time gap is $4$, Wiki's average time gap is $17$, MOOC's average time gap is $3.6$, and GDELT's average time gap is $0.1$. Since baseline methods are relying on RNN and SAM to process the historical temporal information, they implicitly assume the temporal information is "smooth" and with a smaller average time gap. Therefore, baseline methods could potentially work better on the dataset with a smaller average time gap but are less effective on LastFM. GraphMixer does not rely on RNN or SAM, therefore could be less affected by the aforementioned issue.
>
> - **Larger average node degree.** LastFM has a larger average node degree $|\mathcal{E}|/|\mathcal{V}|$ than other datasets, which is potentially prone to over-smoothing (aggregating features from many neighbors makes output representation less distinguishable [1]) and over-squashing effect (aggregating much information into a limited memory might compress too much useful information [2]). For example, according to the dataset statistics in Table 5, LastFM has an average node degree of $653$, which is larger than Reddit's average node degree of $61$, Wiki's average node degree of $17$, MOOC's average node degree $57$, and GDELT's average node degree $216$. Existing methods either use the memory cell in RNN to store temporal information or use SAM to aggregate temporal information from multi-hops, which could be less ideal on a dense graph due to the over-smoothing issues and the over-squashing issues. However, GraphMixer relies less on the aggregation schema, therefore its performance is better than the baseline methods.
>
>     [1] Deeper Insights into Graph Convolutional Networks for Semi-Supervised Learning https://arxiv.org/abs/1801.07606
>
>     [2] On the Bottleneck of Graph Neural Networks and its Practical Implications https://arxiv.org/abs/2006.05205
>
>
> - **Larger maximum timestamp size.** Baselines use a trainable time-encoder, which suffers from an unbounded gradient issue (as discussed in Table 2 and Section 4.2). Recall that the gradient norm when using trainable time-encoder scales proportionally to the maximum timestamp size. Since the maximum timestamp $t_{\max}$ in LastFM is larger than other datasets, the trainable time-encoder is more affected by the unbounded gradient issue. For example, the $t_{\max}$ in LastFM is 137 million, which is significantly larger than the $t_{\max}$ in GDELT (0.2 million), Reddit (2.6 million), Wiki (2.6 million), and MOOC (2.6 million).
>
>
> We would like to thank you again for your constructive feedback and have incorporated them into the Appendix D of the updated manuscript.

---

> > ### Comment · Reviewer_7yc7 · 2022-11-17
> > **Response to the author feedback**
> >
> > I would like to thank the authors for their response to my questions and clarifying. The author has successfully addressed my concerns. it is a good paper.

---

### Official Review · Reviewer_naoZ · 2022-10-25

**Confidence:** 5
**Clarity, Quality, Novelty And Reproducibility:** The paper is clear, novel, and easy t…
**Correctness:** 3
**Technical Novelty And Significance:** 3
**Empirical Novelty And Significance:** 3
**Recommendation:** 8

**Strength And Weaknesses:**

Pros: Focused on the temporal graph learning, the authors propose a conceptually and technically simple architecture GraphMixer for temporal link prediction. GraphMixer not only outperforms all baselines but also enjoys a faster convergence speed and better generalization ability. The author conduct empirical study to identify three key factors that contribute to the success of GraphMixer and highlights the importance of simpler neural architecture and input data structure.

Cons: There is not obvious weakness in the draft.

**Summary Of The Paper:**

In this paper, the authors propose GraphMixer, a conceptually and technically simple architecture that consists of three components: 1 a link-encoder that is only based on multi-layer perceptions (MLP) to summarize the information from temporal links, 2 a node-encoder that
is only based on neighbor mean-pooling to summarize node information, and 3 an MLP-based link classifier that performs link prediction based on the outputs of the encoders. The authors show that GraphMixer attains an outstanding performance on temporal link prediction benchmarks with faster convergence and better generalization performance.

**Summary Of The Review:**

Please refer to the above points.

---

> ### Author Response · Authors · 2022-11-10
> **Response to Reviewer naoZ**
>
> Thank you for acknowledging our article, we believe our work could not only shed some light on the future algorithm design but also inspires future studies to rethink the importance of simpler model architecture.

---

### Official Review · Reviewer_xY6D · 2022-10-31

**Confidence:** 3
**Correctness:** 3
**Technical Novelty And Significance:** 3
**Empirical Novelty And Significance:** 3
**Recommendation:** 6

**Clarity, Quality, Novelty And Reproducibility:**

The work is mostly clear and understandable. Some of the claims around model convergence is a bit misleading and the authors should consider bringing appendix on training speed C.3 into the main paper.

**Strength And Weaknesses:**

Strengths:
- Proposes a novel architecture that achieves performance competitive with other networks without memory or attention mechanisms.
- Strong analysis and ablation of the different design choices made.

Weaknesses:
- The model takes significantly longer to train and there is not significant analysis of the inference cost. This lack of discussion in the main paper also makes some conclusions about faster convergence somewhat misleading.

Questions:
Why are the train average precisions for the competing methods in Figure 3 so much lower than the final results in Table 1. For example, DySAT training AP is around 60-70% while the final eval is 96.71. Are all the results from reimplementations or from taken from other papers.

It would likely be helpful to bring appendix C.3 into the paper and analyze and discuss more about the model efficiency since the model does take significantly longer to train than some baselines like TGN. If some of this is due to preprocessing, it would be helpful to separate the time required for each and/or analyze the inference FLOPs for your model compared to the others.


**Summary Of The Paper:**

In this paper, they carefully design and propose the GraphMixer network for temporal link prediction in dynamic graphs. With a carefully designed and ablate architecture separated into three components, the link-encoder, node-encoder, and MLP-Mixer based link classifier, they are able to achieve high performance on this task that has only previously been attained with RNN and Self-Attention based architectures. They include extensive experiments to study the different architecture choices they made and their effect on performance and generalization.

**Summary Of The Review:**

This work proposes a novel architecture for link prediction in temporal graph networks that achieves very good performance without RNN or self-attention mechanisms. They achieve this using a carefully constructed network with MLP-mixer. They do detailed experiments to analyze and ablate the different design choices they made. A large strength of this paper is the extensive experiments ablating the details of the design choices they made how it affects performance, generalization, and training stability. This greatly helps the community understand how these architecture choices affected their model and may extend to other models. The paper may have somewhat limited novelty since the different parts of the design have all been explored elsewhere and in terms of setting a new SOTA, the model still has some under-analyzed weaknesses such as the slow training speed.

---

> ### Author Response · Authors · 2022-11-10
> **Response to Reviewer xY6D (Part 1)**
>
> **Q1. The model takes longer to train. Suggest explicitly calculating the computation time and bringing appendix C3 into the paper**
>
>
> A1. Thank you for the valuable feedback and suggestions. As suggested by the reviewer, we explicitly calculate the computation time (i.e., the forward and backward propagation time) and report the results in the last row of the following table.
> As shown in the table, the computation time of GraphMixer is very close to or even faster than most baseline methods.
> We update our discussion in Appendix C3 and update the code in [anonomous repository](https://anonymous.4open.science/r/GraphMixer/link_pred_train_utils.py) (might need to clear the browser cache to see the update) to explicitly calculate the computation time. Due to the space limit, we decided to temporally keep Appendix C3 in the appendix but added a pointer in Section 4.1 to our discussion on wall-clock time in Appendix C3.
>
> |            |    Reddit   |    Wiki   |    MOOC   |    LastFM   |    GDELT    |
> |------------|:-----------:|:---------:|:---------:|:-----------:|:-----------:|
> | JODIE      |   $5$ sec   |  $2$ sec  |  $4$ sec  |   $11$ sec  |   $16$ sec  |
> | DySAT      |   $33$ sec  |  $6$ sec  |  $16$ sec |   $41$ sec  |   $83$ sec  |
> | TGAT       |   $15$ sec  |  $4$ sec  |  $8$ sec  |   $28$ sec  |   $41$ sec  |
> | TGN        |   $8$ sec   |  $2$ sec  |  $5$ sec  |   $15$ sec  |   $32$ sec  |
> | APAN       |   $13$ sec  |  $4$ sec  |  $9$ sec  |   $28$ sec  |   $25$ sec  |
> | CAWs-mean  | $1,893$ sec | $277$ sec | $641$ sec |   $1,797$ sec  | $4,544$ sec |
> | CAWs-attn  | $1,930$ sec | $282$ sec | $653$ sec | $1,832$ sec |   $4,634$ sec  |
> | TGSRec     |  $538$ sec  | $157$ sec | $656$ sec | $1,810$ sec | $3,707$ sec |
> | GraphMixer |   $12$ sec  |  $3$ sec  |  $7$ sec  |   $21$ sec  |   $32$ sec  |
>
> The time reported in the original table (Table 7 in Appendix C3) considers both the data pre-processing time (e.g., sorting nodes in subgraph according to temporal order and removing duplicated edges) and the computation time, where data pre-processing time is longer than the computation time. For example, the data pre-processing takes 41 seconds on Reddit, 9 seconds on Wiki, 20 seconds on MOOC, 48 seconds on LastFM, and 71 seconds on GDELT. However, by caching the pre-processed data (i.e., the model input variable `inputs` in [link\_pred\_train\_utils.py](https://anonymous.4open.science/r/GraphMixer/link_pred_train_utils.py)), we only need to prepossess data once at the beginning of the training process because our neighbor selection is deterministic and the input data does not change at each epoch.
>
> Besides, we would like to point out that we choose to compare the wall-clock time instead of the FLOPs because computing FLOPs is non-trivial for baseline methods, especially when they are not implemented with pure PyTorch functions. Please refer to [this forum](https://discuss.pytorch.org/t/correct-way-to-calculate-flops-in-model/67198/8) for more details on why computing exact FLOPs is non-trivial for general neural network architectures.
>
> **Q2: Are results in Table 1 re-implementation or from existing paper? Why training precision is lower than the testing precision results in Table 1?**
>
> A2: Thank you for bringing up this point. All experiment results reported in this paper are obtained by re-running the official/publicly released implementations. In the near future, we would also like to clean up and release the code for all baseline implementations, including the functions we implemented on top of the baseline methods to gather important information for model analysis and plotting.
>
> The testing score in temporal graph learning is higher than the training score because
> - the chronological splits for the training/validation/testing sets give a model with more historical graphs to use during testing than training,
> - the performance of baselines is less stable than ours.
>
> More specifically, when splitting training/validation/testing sets by chronological order, testing set data has more recent timestamps than the training set data. In temporal graph learning, since a model is allowed to have access to all historical graphs before its current timestamp, more historical graphs are allowed to use during the testing phase than the training phase when doing prediction. As a result, the testing score is usually higher than the training score in temporal graph learning, even though the models are not well trained during the training phase because they just need to make predictions based on historical graphs. Meanwhile, since GraphMixer has a relatively more stable performance due to its simplicity, GraphMixer is less affected by the aforementioned chronological order split issue and could maintain a stable performance both during training and testing.

---

> > ### Author Response · Authors · 2022-11-10
> > **Response to Reviewer xY6D (Part 2)**
> >
> > We would like to thank you again for your constructive feedback and are glad to incorporate the above discussion into the updated draft. Since our work shows conceptually and technically simple methods are enough to achieve SOTA performance on temporal graph learning, we believe our work could not only shed some light on the future algorithm design for temporal graph learning but also inspires future studies to rethink the importance of simple model architectures than a complicated one.

---

> > > ### Comment · Reviewer_xY6D · 2022-11-16
> > > **Followup questions**
> > >
> > > I would like to thank the authors for their response to my questions and clarifying. I have a few last questions about the results.
> > >
> > > Do you intend to completely remove the information in the appendix on the preprocessing cost?  I believe since a major advantage of this paper is the simplicity and speed, that there should be more not less discussion of this. How much does the preprocessing cost compared to the other methods? Would this limit it's practical use or could it be optimized during inference.
> > >
> > > It still just isn't clear to me how much of the improvements is due to the MLP-Mixer. I think the paper would be clearer with less focus on the simplicity aspect. The paper achieves strong performance due to the many design choices which were ablated in this work. Compared to self-attention, while you demonstrated some reasons the MLP-Mixer may achieve better performance, it isn't clear that issues like being unable to distinguish $[a_1,a_1]$ and $[a_1]$ couldn't be solved with some basic encoding changes. It isn't clear to me that a simple RNN or even self-attention architecture is inherently significantly more complicated since you also have to be quite careful in the design of the input and output encodings. Are there more empirical measures to distinguish the architecture?

---

> > > > ### Author Response · Authors · 2022-11-17
> > > > **Response to Reviewer xY6D (Part 1)**
> > > >
> > > > **Q1. Please keep the information in the appendix on the preprocessing cost.**
> > > >
> > > > A1. Thank you for your suggestion. We totally agree that the discussion on pre-processing time is important. Therefore, we keep the data pre-processing time at the end of Appendix C3 (after Table 8) and use one paragraph to highlight our pre-processing time (which includes sorting the nodes according to timestamps). By doing so, we can focus on the computation time in Table 7 to avoid confusion.
> > > > Please notice that although the overall time of our method might be longer than some of the baselines, our method can achieve the best model performance without sacrificing the overall time.
> > > >
> > > > Meanwhile, we would like to clarify that the simplicity we mentioned does not necessarily mean fast computation time. Instead, we are mainly referring to whether the methodology is conceptually and technically simpler. Please refer to the more detailed justification of our response in Q3.
> > > >
> > > > **Q2. How many of the improvements are due to the MLP-Mixer? It isn't clear that issues like "being unable to distinguish" couldn't be solved with some basic encoding changes.**
> > > >
> > > > A2. Thank you for the question. To understand how MLP-mixer affects model performance, we conduct the ablation study in Section 4.3 by replacing the MLP-mixer with SAM, where the time-encoding services the same functionality as the positional encoding. In particular, we show that self-attention with sum-pooling could already solve the indistinguishable issue and achieve the same training accuracy as MLP-mixer in Figure 7. However, using MLP-mixer has better overall performance because it has fewer parameters and could generalize better.
> > > >
> > > > **Q3.  I think the paper would be clearer with less focus on the simplicity aspect.**
> > > >
> > > > A3. Thank you for your suggestion.
> > > > Please notice that our "simplicity" does not just simply refers to whether a function has faster computation time than another function, but mainly refers to whether the methodology is conceptually and technically simpler than another one. For example,
> > > >
> > > > - JODIE, TGN, and CAW maintain a memory block for each node and use RNN to update the memory upon each interaction between nodes. The conceptual idea and implementation techniques behind using memory block and RNN to store the historical interaction information and update the information using RNN are more complicated.
> > > >
> > > > - Similarly, TGAT and TGSRec are conceptually more complicated than ours since they additionally introduce the concept of self-attention for temporal neighbor aggregation.
> > > >
> > > > However, in this paper, we found that although the aforementioned designs could lead to a good performance, in practice neither of them is always necessary. In practice, we can achieve SOTA by using a mean-pooling node-encoder to aggregate node features and an MLP-based link-encoder to aggregate link features, without requiring the aforementioned algorithm design.
> > > > Most importantly, thanks to the simplicity of our method, we can achieve a deeper understanding of the behavior of temporal graph networks from convergence, generalization, and optimization landscape. For example, we found that the trainable time-encoder used in previous works suffers from exploding gradient and hurt the optimization landscape. We believe this paper could greatly help the community understand how these architectural choices affected their model and may extend to other models.

---

> > > > > ### Comment · Reviewer_xY6D · 2022-11-21
> > > > > **Increasing Score**
> > > > >
> > > > > I would like to thank the authors for their response to my questions. Since my main concern about the preprocessing has been addressed I will increase my score and recommend acceptance of this paper. I agree with the other reviewers that they demonstrated and analyzed and ablated a good relatively simple architecture that does well on this temporal problem with the caveat that I am less familiar with this particular temporal graph problem, and am less able to judge the significance of their empirical results.

---

> > > > ### Author Response · Authors · 2022-11-17
> > > > **Response to Reviewer xY6D (Part 2)**
> > > >
> > > > **Q4. Why RNN and SAM are more complicated than MLP? Any empirical measures to distinguish the architecture?**
> > > >
> > > > A4. This is indeed an interesting question. We believe RNN or SAM are generally considered more complicated because
> > > >
> > > > - RNN has two different inputs (one for historical input and another one for new input) and takes the recurrent structure of the input data into consideration.
> > > > The most common way of using RNN in temporal graph learning is to maintain a memory block for each node and update the memory block via RNN upon each interaction happens (e.g., JODIE and TGN). Maintaining the memory block when using RNN is complicated and tracking the influence on model performance is non-trivial because the memory is updated upon each interaction happens.
> > > > If not using the memory block, we have to feed a long sequence of inputs recursively into RNN, which is also more complicated than using MLP.
> > > > Since our method is not an RNN-based method, we only compare the number of parameters of our model to RNN-based baselines in Table 8, but do not explicitly conduct other ablation studies on RNN.
> > > >
> > > > - SAM uses a self-attention mechanism for information aggregation, which can be thought of as an advanced module that is built upon MLP that takes the inner dependency between data into consideration. We explicitly compare with SAM-based aggregation methods in Section 4.3 and highlight the advantage of using MLP-based aggregation over SAM.
> > > >
> > > > In terms of the ablation study between RNN and SAM,  we would like to note that the performance comparison between JODIE, TGAT, and TGN can be considered an ablation study between RNN and SAM. This is because TGN is a combination of the RNN module from JODIE and the SAM module from TGAT.
> > > >
> > > > ==================================================
> > > >
> > > > Once again, thanks for your time and constructive comments. We would be happy to elaborate more if further clarification is needed.

---

### Decision · Program_Chairs · 2023-01-20

**Decision:**

Accept: notable-top-5%

**Justification For Why Not Higher Score:**

N/A

**Justification For Why Not Lower Score:**

The authors present a simple and elegant solution that achieves state-of-the-art performance on temporal graph learning-related tasks. The proposed solution has the potential to open the door to research for simpler, more interpretable, yet effective solutions.

**Metareview: Summary, Strengths And Weaknesses:**

I Summary:

- I.1 Investigated Problem:
The paper investigates if memory and self-attention mechanisms are indispensable for temporal graph learning. Motivated by the fact that it is non-trivial to understand which parts of the self-attention and memory-based models contribute to its success and whether these components are indispensable, the paper identifies 3 key factors to a simpler and elegant solution named GraphMixer.

- I.2 Proposed Solution: GraphMixer is conceptually and technically a simpler architecture for temporal link prediction that achieves:
   - Better generalization compared to all tested baselines;
   - Faster convergence speed.

- I.3 Validity Proof of the Proposed Solution:
Illustrative and intuitive visualization as well as an empirical piece of evidence is provided to support the validity of the proposed solution.
Moreover, an extensive study identifying the following 3 key factors contributing to the success of GraphMixer is conducted:
    - The use of a fixed time-encoding function:
        - The authors demonstrated that a trainable time-encoding function could cause instability during training which leads to a non-smooth landscape objective;
    - The use of MLP-Mixer in GraphMixer’s Link Encoder is a good alternative to self-attention and its simplicity leads to a better generalization performance;
    - Using a simple input structure that is better aligned with their labels allows a simple neural network model to capture the underlying mapping between the input and labels which leads to a better generalization.

II Strengths:

- II.1 From a structural point of view:
    - The authors formulated the problem investigated in terms of questions and presented the content of the paper as answers to these questions;
    - Used circled numbers and bullet points in the presentation of the proposed solution as well as question and answer type of analysis for the experiments. This is an example of a very well-written and structured paper showcasing the simplicity and effectiveness of a novel method.

II.2 From an analytical point of view:
 - The clarity of the motivation: Presented in terms of questions;
 - The presentation of the designed method: Simple and elegant presentation for a simple and elegant solution;
 - The visual and empirical evidence provided and the analysis performed to identify the key factors to the performance of the method provide good intuition about the effectiveness of the proposed solution more intuitive to understand;
 - Comparisons are conducted with several features of the existing methods. In fact, it has been shown that the presented solution is comparable to or better than most baselines;
 - The transparency aspect of the submission as open-source code is provided for reproducibility purposes and the authors are planning to clean up and release the code for all baseline implementations to gather important information for model analysis and plotting.

- II.3 From a perspective of soundness (development, unity, and coherence) and completeness (correctness):
  - The strength points mentioned above are sufficient evidence of the soundness and completeness of the paper.

III Addressing what can be thought of as weaknesses:

As has been mentioned by most reviewers, there isn’t an obvious weakness to mention. Most of what could be thought of as weaknesses have been addressed and clarified by the authors during the rebuttal period. Unanimously, the reviewers agree on the acceptance of the submission.

IV. Potential of the paper:

- IV.1 From a Potential perspective (Potential of the paper to the community): The proposed solution has great potential to be of benefit to the whole community, As mentioned by the authors in the conclusion, not limited to temporal graph learning and to name a few, an interesting direction would be to design algorithms that could automatically select the best input data and data pre-processing strategies for different downstream tasks.

**Note From Pc:**

if the above contains the word "oral" or "spotlight" please see: "oral" presentation means -> notable-top-5% and "spotlight" means -> notable-top-25%. As stated in our emails, we are disassociating presentation type from AC recommendations